# Optogenetic frequency scrambling of hippocampal theta oscillations dissociates working memory retrieval from hippocampal spatiotemporal codes

Guillaume Etter [1] ✉, Suzanne van der Veldt[1], Jisoo Choi[1] & Sylvain Williams [1] ✉

The precise temporal coordination of activity in the brain is thought to be fundamental for memory function. Inhibitory neurons in the medial septum provide a prominent source of innervation to the hippocampus and play a major role in controlling hippocampal theta (~8 Hz) oscillations. While pharmacological inhibition of medial septal neurons is known to disrupt memory, the exact role of septal inhibitory neurons in regulating hippocampal representations and memory is not fully understood. Here, we dissociate the role of theta rhythms in spatiotemporal coding and memory using an all-optical interrogation and recording approach. We find that optogenetic frequency scrambling stimulations abolish theta oscillations and modulate a portion of neurons in the hippocampus. Such stimulation decreased episodic and working memory retrieval while leaving hippocampal spatiotemporal codes intact. Our study suggests that theta rhythms play an essential role in memory but may not be necessary for hippocampal spatiotemporal codes.

The precise temporal coordination of neuronal activity is thought to be fundamental for memory encoding and retrieval. In particular, the medial septum (MS) has been proposed to act as the main zeitgeber to downstream structures and provides the largest subcortical inputs to the hippocampus[1]. The MS is a neurochemically heterogeneous structure composed of GABAergic[2], mainly parvalbumin (PV)-positive neurons[3], along with cholinergic[4] and a smaller population of glutamatergic neurons[5,6]. MS PV cells project directly onto GABAergic interneurons in the hippocampus giving rise to feedforward inhibitory control of hippocampal pyramidal cells[3]. Additionally, PV interneurons within the hippocampus are essential in pacing theta (~8 Hz) rhythms[7], and optogenetic stimulation of MS PV neurons directly[8,9] or of their terminals[10] in the hippocampus are associated with frequency-specific pacing of hippocampal oscillations, whereas inhibition of the MS in vivo has been associated with reduced theta oscillation power[11–13].

While complete MS optogenetic inhibition has been associated with spatial memory impairments[14], these effects could potentially be attributed to disruption of cholinergic functions[15,16], which are known

to be critical for memory. More recently, the activity of non-cholinergic MS PV cells has been shown to be necessary for memory encoding and retrieval[17,18]. Notably, the hippocampus is also a core structure for episodic[19,20] and working[21] memory. Since the MS is essential in generating and maintaining hippocampal theta rhythms, disruption of MS activity is likely to impact downstream hippocampal physiology and working memory[22]. Although the exact physiological mechanisms of hippocampus-dependent memory are currently unknown, it has been proposed that hippocampal place cells[23] that encode specific locations of a given context could support episodic memory[24]. In the hippocampal subfield CA1, spatial tuning depends on contextual sensory inputs[25]. Other variables such as time and distance can also be encoded during visually guided locomotion[26,27] but also in the absence of sensory cues[28], likely using internal information including self-motion[29] (see Mehta[30] and McNaughton[31] for review). Representations of time and space can be represented conjunctively in hippocampal neurons, and such multiplexed spatiotemporal codes could be a candidate substrate for working memory[32–34]. In addition to

[1]McGill University & Douglas Mental Health University Institute, Montreal, Canada. ✉e-mail: etterguillaume@gmail.com; sylvain.williams@mcgill.ca

clamping distal visual cues using a treadmill[35–37] or using virtual reality paradigms[26,27,38], spatiotemporal codes have also been extracted analytically using generalized linear models that implement space, time, and distance[37]. However, such approaches have not been employed extensively on recordings of neuronal activity during free exploration.

Several studies suggest that hippocampal theta rhythms could underlie temporal codes since theta rhythms tightly orchestrate hippocampal activity[39,40]. While time cells have also been reported in both CA1 and CA3 of rodents performing tasks that do not require working memory[41], pharmacological inhibition of the MS results in specific disruption of time but not place cells and is associated with decreased working memory[36]. An important drawback of pharmacological approaches is that they do not distinguish the relative contribution of GABAergic versus cholinergic cells to memory function[42]. Notably, inhibition of MS cholinergic activity was found to alter hippocampal spatial representations[43] and decrease working memory performance[44,45]. Surprisingly, pharmacological inhibition of the MS was associated with reduced theta oscillation power but not place fields[46], and this resilience of place cell activity during diminished theta was not due to experience-related plasticity mechanisms[47]. Previous attempts at inhibiting MS GABAergic interneurons specifically using optogenetics, were associated with only a partial reduction, but not complete disruption of theta signals[13]. In turn, optogenetic pacing of theta oscillation has only been associated with minor changes in place cell characteristics, including a slight shift in firing frequency[9] and phase[48]. Additionally, while it is hypothesized that MS inputs could directly control hippocampal temporal codes, causal evidence is still lacking. To this day, the exact role of MS-PV neurons in orchestrating hippocampal spatiotemporal codes and memory remains unknown.

Here, we controlled MS PV activity using optogenetics to pace or abolish theta oscillations using a red-shifted excitatory opsin. We propose an approach to completely abolish hippocampal theta rhythms based on optogenetic frequency scrambling stimulations of MS neurons. Alternatively, pacing theta rhythms at their natural frequency in the same animals provides within-subject controls. We combined optogenetic control with calcium imaging of CA1 pyramidal cells in mice running on a linear track with sequential tones. In these conditions, we could separate place, time, and distance cells using an information theoretic approach. When performing optogenetic frequency scrambling of theta signals, both place and time representations were preserved and only a small subset of CA1 pyramidal cells was modulated by stimulation. We next found that ablation of theta oscillation was associated with impaired working memory retrieval suggesting that MS PV cells play a critical role in generating hippocampal theta oscillations that are necessary for memory retrieval but are not involved in spatiotemporal representations.

## Results

### CA1 pyramidal cells encode spatiotemporal information

To examine the spatiotemporal codes in large populations of CA1 principal cells, we injected a viral vector expressing GCaMP6$_{fast}$ under a CamKII promoter in the CA1 region of the hippocampus, implanted a GRIN lens above the injection site, and performed calcium imaging recordings of pyramidal neurons using open-source miniscopes[49,50] (Fig. 1a, b; see Supplementary Fig. 1 for detailed histology). We extracted spatial footprints of neurons (Fig. 1c) and their corresponding calcium transients (Fig. 1d) using CNMFe[51]. To tease apart the spatial and temporal tuning properties of principal cells, we developed a task combining a linear track with three-tone cues triggered by motion sensors at both ends of the track. A new tone was instantly triggered at the end of each run, informing the mice of their progression toward reward delivery. Every fourth run was cued with a high-pitched, continuous tone that was associated with the delivery of a reward at the end of the linear track (Fig. 1e). The absolute location of each mouse, along with the time elapsed and distance traveled since

the departure from the reward site were monitored (Fig. 1f). Using these variables and binarized neuronal activity, we computed probabilistic tuning curves (Fig. 1g) and derived mutual information (MI) between neuronal activity and location, time, as well as the distance for each recorded cell. In contrast to correlation-based analyses, MI does not assume linear, monotonic relationships between neuronal activity and behavioral variables but rather expresses the amount of uncertainty of one variable that can be explained by the other. The significance of MI values was tested using shuffled surrogates that underwent circular permutations ($n = 1000$) to preserve the temporal dynamics of calcium transients. Neurons that encoded exclusively one variable with a MI greater than shuffled surrogates 95% of the time ($p \leq 0.05$) were labeled as either place-modulated (spatial), time-modulated (temporal), or distance-modulated (see Methods). For the following analyses, we focused on candidate cells that only encoded one significant variable (Fig. 1h). Importantly, time-modulated cells were not systematically active at particular locations, and place-modulated cells were not systematically active at a given time (Fig. 1i, j). While the majority of cells encoding a single variable were place-modulated, a large portion of neurons were conjunctive neurons that encoded more than one variable ($17.98 \pm 1.77\%$). In contrast, cells encoding place exclusively represented $9.02 \pm 1.61\%$ of the total recorded population, while $1.79 \pm 0.68\%$ selectively encoded distance and $1.27 \pm 0.09\%$ selectively encoded time (Fig. 1k; additional examples of neurons tuned to time, space, or distance along with their information content are shown in Supplementary Fig. 2).

Although our information-theoretic approach can disentangle overlapping variables by isolating cells that only significantly encode one variable, we further tested the relevance of each cell type in encoding spatiotemporal variables using a naive Bayesian classifier to decode location (Fig. 1l–n), the time elapsed (Fig. 1o–q), and distance traveled (Fig. 1r–t) on the linear track[52]. We estimated the current state of each mouse by computing the maximum *a posteriori* (MAP) value given neuronal activity and bootstrapped tuning curves computed using either actual or circularly shuffled binarized activity (Fig. 1l, o, r; see Methods for detailed protocol). The quality of predictions was assessed using confusion matrices (Fig. 1m, p, s) and by computing the Euclidean distance between the predicted state and the actual state (Fig. 1n, q, t). Importantly, our Bayesian decoder yielded an average error of 16.58 cm, which was significantly lower than when decoding from shuffled surrogates (50.95 cm; paired *t*-test, $t_4 = 19.75$, $p \leq 0.0001$), and decoding using spatially modulated cells was significantly more accurate than when using non-spatially modulated cells (paired *t*-test, $t_4 = 34.54$, $p \leq 0.0001$; Fig. 1n). Similarly, the average decoding error for time elapsed was 6.45 s, which was significantly lower than error computed using shuffled surrogates (19.12 s; paired *t*-test, $t_4 = 18.01$, $p \leq 0.0001$). Decoding using time-modulated cells yielded significantly better accuracy compared to non-time-modulated cells (paired *t*-test, $t_4 = 3.163$, $p = 0.0341$; Fig. 1q). Finally, the average distance error using our decoder was 61.40 cm, which was significantly lower than that of shuffled surrogates (189.6 cm; *t*-test, $t_4 = 28.79$, $p \leq 0.0001$). Decoding using distance-modulated cells yielded significantly lower errors compared to non-distance-modulated cells (*t*-test, $t_4 = 4.595$, $p = 0.0101$; Fig. 1t).

### Selective MS optogenetic control of theta oscillations

To examine the relative contribution of MS-generated theta signals to hippocampal spatiotemporal codes, we transfected the red-shifted excitatory opsin ChrimsonR in the MS (Fig. 2a). In contrast to inhibitory opsins, ChrimsonR allowed us to either scramble or pace theta signals within subjects. Additionally, ChrimsonR is more effective than the more widely used Channelrhodopsin-2 and provides the capability to combine optogenetics with calcium imaging[53]. $14.91 \pm 3.57\%$ of PV cells

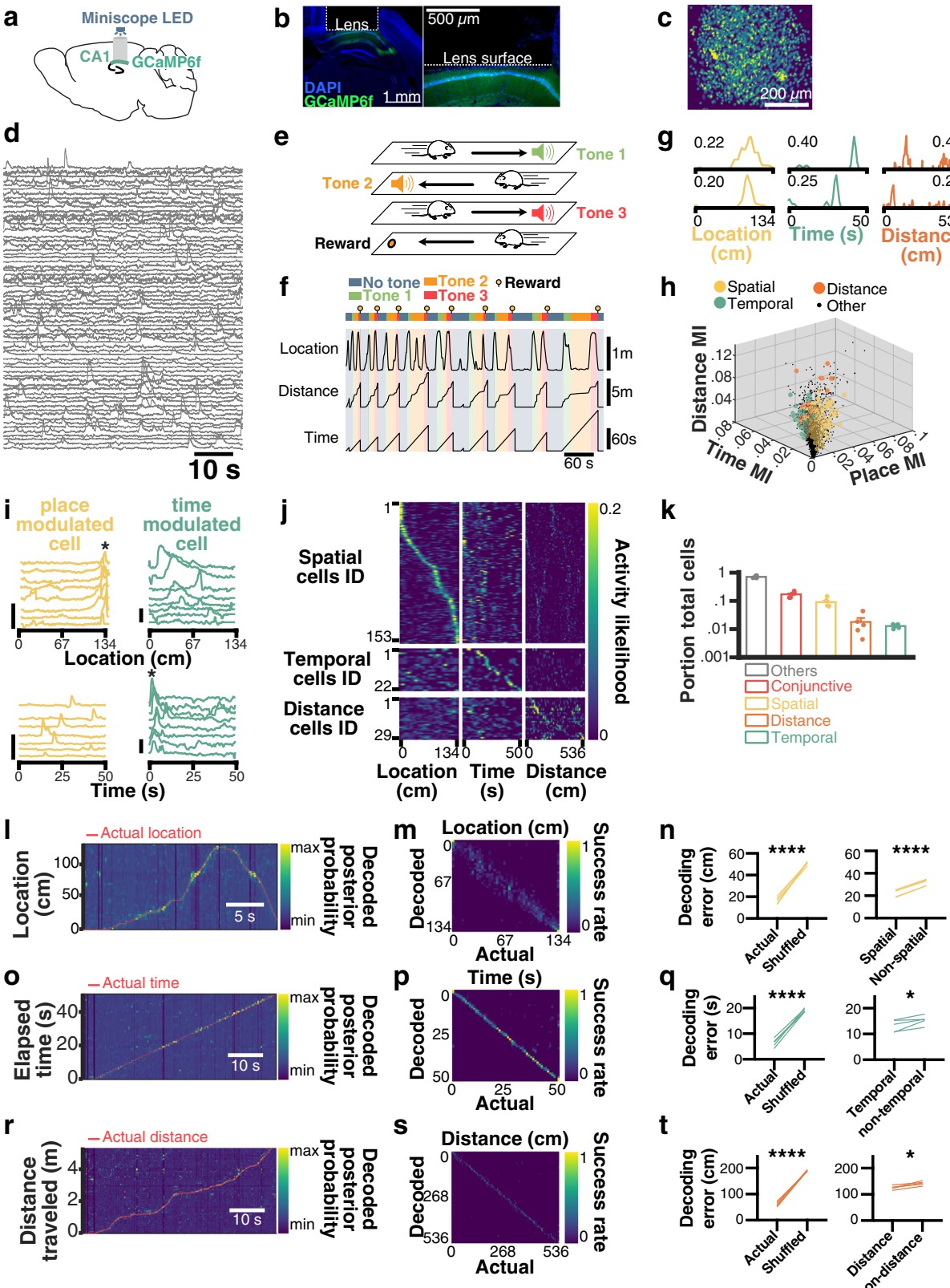

expressed ChrimsonR, which we found sufficient to exert pervasive control over hippocampal oscillations (*n* = 4 mice; Fig. 2b, c). In contrast, we found virtually no expression of ChrimsonR in ChAT cells (1.05 ± 1.052% of ChAT cells also expressed ChrimsonR; Fig. 2d, e).

We then implanted mice with fiber optics over the MS and performed 638 nm laser stimulation while recording local field potentials (LFP) in CA1 (Fig. 2f). We found that MS scrambled and 8 Hz optogenetic stimulation could disrupt or pace theta oscillations, respectively (Fig. 2g). While baseline natural theta displays some frequency variability in the 4–12 Hz frequency band, 8 Hz stimulations led to consistent and stable hippocampal oscillations at that frequency, whereas scrambled stimulations consistently abolished

**Fig. 1 | Spatial and temporal codes are present in partially overlapping populations of CA1 neurons. a** Mice were injected with GCaMP6f in hippocampal subfield CA1 and subsequently implanted with a GRIN lens over the pyramidal cell layer for miniscope imaging. **b** Post-mortem coronal section from one example mouse. blue, DAPI; green, GCaMP6f. **c** Planar projection of extracted spatial footprints from one example mouse (blue, background; yellow, footprints). **d** Corresponding calcium transients from a portion of recorded cells. **e** Behavioral paradigm used to disentangle time, place, and distance coding cells (see Methods). **f** Behavioral outcomes for one example mouse. **g** Example pairs of neurons modulated by place (yellow), time (green), or distance (orange) along with corresponding peak activity likelihood. **h,** 3D scatter plot of the distribution of cells selectively encodes a single variable along with their associated mutual information (MI) in each dimension. Cells in black either encode more than one variable or did not encode any variable significantly more than shuffled surrogates. **i** Individual traces from one example place-modulated cell (yellow) and one example time-

modulated cell (green). Vertical scale bars: 1$\Delta F/F$. **j** Tuning curves for a place, time, and distance cells, sorted by the location of peak activity likelihood for each dimension (location, time, and distance; dark blue, low likelihood; bright yellow, high likelihood). **k** Portion of total cells (log scale) encoding for one, several, or no variables ($N$ = 5 mice). Bayesian decoding of location (**l–n**), time (**o–q**), and distance (**r–t**; $N$ = 5 mice). For each encoded variable, posterior probabilities are plotted with the corresponding actual state (**l, o, r**; dark blue, low probability; bright yellow, high probability). Confusion matrices between the actual vs decoded values (**m, p, s**; dark blue, low rate; bright yellow, high rate). Decoding errors using all cells vs shuffled surrogates (**n, q, t**, left panels). Right panels: **n** decoding errors using spatial cells vs nonspatial cells ($t_4$ = 34.54, $p \leq 0.0001$). **q** Decoding errors using temporal cells vs. nontemporal cells ($t_4$ = 3.163, $p$ = 0.034). **t** Decoding errors using distance vs. non-distance cells ($t_4$ = 4.595, $p$ = 0.0101). Paired two-tailed $t$-tests were used in all panels of (**n, q, t**). *$p \leq 0.05$; ****$p \leq 0.0001$. All bar plots represent the mean ± SEM of at least two independent experiments.

theta rhythms (Fig. 2h). We found that oscillation strength (OS) in the theta band (see Methods) was significantly decreased by scrambled stimulations (0.45 ± 0.01) compared to baseline epochs (0.67 ± 0.01, $p \leq 0.0001$) and were not significantly different from OS of the white noise control signal (0.50 ± 0.01, $p$ = 0.99). On the other hand, 8 Hz stimulations increased theta power significantly (0.82 ± 0.01, $p \leq 0.0001$; $n$ = 59 epochs; Fig. 2i) compared to scrambled stimulations. We also found a significant interaction between our stimulation patterns and the LFP frequency band ($F_{10}$ = 6.467, $p \leq 0.0001$). In particular, scrambled frequency stimulation significantly decreased theta power (0.341 ± 0.06 portion of baseline theta band power; $p$ = 0.0394, pairwise $t$-test), while 8 Hz stimulations significantly increased theta power (3.302 ± 0.76 portion of baseline theta band power, $p$ = 0.0004, pairwise $t$-test; Fig. 2j) leaving other frequency bands unaltered.

While our calcium imaging and electrophysiological analyses only included periods of locomotion (see Methods), we also found that we were able to reliably abolish (Supplementary Fig. 3a, b) or pace (Supplementary Fig. 3a, c) theta oscillations regardless of locomotor state (including periods of restfulness). While natural theta OS is correlated to locomotor speed (Pearson $R^2$ = 0.059, $p$ = 0.001; $n$ = 179 independent epochs; Supplementary Fig. 3d), abolishing theta led to a loss of such correlation (Pearson $R^2$ = 0.008, $p$ = 0.223; $n$ = 179 independent epochs; Supplementary Fig. 3e), as did 8 Hz stimulations (Pearson $R^2$ = 0.0008, $p$ = 0.714; $n$ = 177 independent epochs; Supplementary Fig. 3f) suggesting that locomotor states did not override the effects of optogenetic stimulations on theta oscillations.

Hippocampal sharp-wave ripples (SWRs) play an essential role in memory consolidation[54–56] and stimulation of MS cholinergic neurons has been associated with reduced ripple activity[57] and impaired working memory[45]. Although we found virtually no expression of ChrimsonR in MS cholinergic neurons, it was essential to measure the impact of our MS optogenetic stimulation on ripple physiology. To this end, we recorded CA1-LFP and performed 5s ON, 5s OFF scrambled optogenetic stimulation in freely behaving mice exploring an open field (Supplementary Fig. 4a). We measured attributes of ripple events before and during scrambled stimulation, and found no changes in power (unpaired, two-tailed $t$-test, $t_6$ = 0.076, $p$ = 0.941; Supplementary Fig. 4b, left panel), frequency of occurrence (unpaired, two-tailed $t$-test, $t_6$ = −1.688, $p$ = 0.142; Supplementary Fig. 4c, left panel), or width (unpaired, two-tailed $t$-test, $t_6$ = 0.124, $p$ = 0.905; Supplementary Fig. 4d, left panel). Similarly, applying 8 Hz optogenetic stimulation had no discernable effects on ripple power (unpaired, two-tailed $t$-test, $t_6$ = 0.378, $p$ = 0.718; Supplementary Fig. 4b, right panel), frequency (unpaired, two-tailed $t$-test, $t_6$ = −1.643, $p$ = 0.151; Supplementary Fig. 4c, right panel), and width (unpaired, two-tailed $t$-test, $t_6$ = −0.138, $p$ = 0.894; Supplementary Fig. 4d, right panel). Together with our histological results and a previous report that optogenetic stimulation of MS cholinergic neurons reduces the occurrence of SWRs[57], we find

that these optogenetic stimulations are not affecting cholinergic inputs to the hippocampus.

## Combining MS optogenetic stimulation and hippocampal calcium imaging

To study the effects of theta manipulations on hippocampal spatial and temporal codes, we combined MS optogenetic stimulation with calcium imaging in CA1. This experimental paradigm raises two potentially important issues: GRIN lens implants involve tissue damage that could alter the physiological state of theta oscillations, and the wavelength spectrum of the excitation LED used to excite GCaMP6f could potentially overlap with that of the opsin located on the terminal fibers of GABAergic MS fibers in the hippocampus.

To verify whether GRIN lens implants altered the physiology of theta, we first implanted mice with a GRIN lens in the right hippocampus and two electrodes in both left and right hippocampi, and found comparable theta signals in both hemispheres (Fig. 3a). We compared theta oscillations during open field exploration in mice with both a GRIN lens and an attached LFP electrode, to mice with electrodes only, and found no significant differences between both groups (Fig. 3b). We did not find any significant difference between relative theta power in mice implanted with a GRIN lens and LFP electrode (0.14 ± 0.007) versus an electrode only (0.154 ± 0.008; $t$-test, $t_{54}$ = 1.060, $p$ = 0.29).

Although previous reports have described that combining optogenetic stimulation of ChrimsonR in cell bodies with imaging of GCaMP in neurons at the vicinity of terminals is possible with minimal crosstalk[53], we next monitored any potential opsin activation of MS terminals in the hippocampus by recording CA1-LFP while emitting excitation light with our miniscope through a GRIN lens. After calibrating miniscope light output power (Fig. 3c), we found no effect of the miniscopes blue excitation light on endogenous theta power (1ANOVA, $F_{(4,295)}$ = 0.7729, $p$ = 0.5435; Fig. 3d). Since it has been reported that miniscope excitation light can induce a slight depolarization of terminals transfected with ChrimsonR[53], potentially hindering further optogenetic-induced depolarization, we next applied MS optogenetic stimulations while imaging with a -0.3 mW/mm² miniscope LED power and were able to significantly disrupt or pace theta using scrambled or 8 Hz stimulation respectively (Friedman test $\chi^2$ = 6.000, $p$ = 0.0278; Fig. 3e).

## Disruption of theta rhythms modulates a small portion of CA1 cells

We then performed phasic (5 s ON, 5 s OFF) optogenetic stimulations of MS while recording CA1 pyramidal cells as mice freely explored an open field (Fig. 4a). We found that a portion of recorded cells were consistently excited in these conditions, while others were inhibited (Fig. 4b; see Methods). The activity of pyramidal cells during stimulations was overall lower compared to baseline, for both scrambled stimulation during running (Pearson correlation, $R^2$ = 0.567, $p \leq 0.0001$) and rest ($R^2$ = 0.521, $p \leq 0.0001$) periods, as well as for 8 Hz stimulation

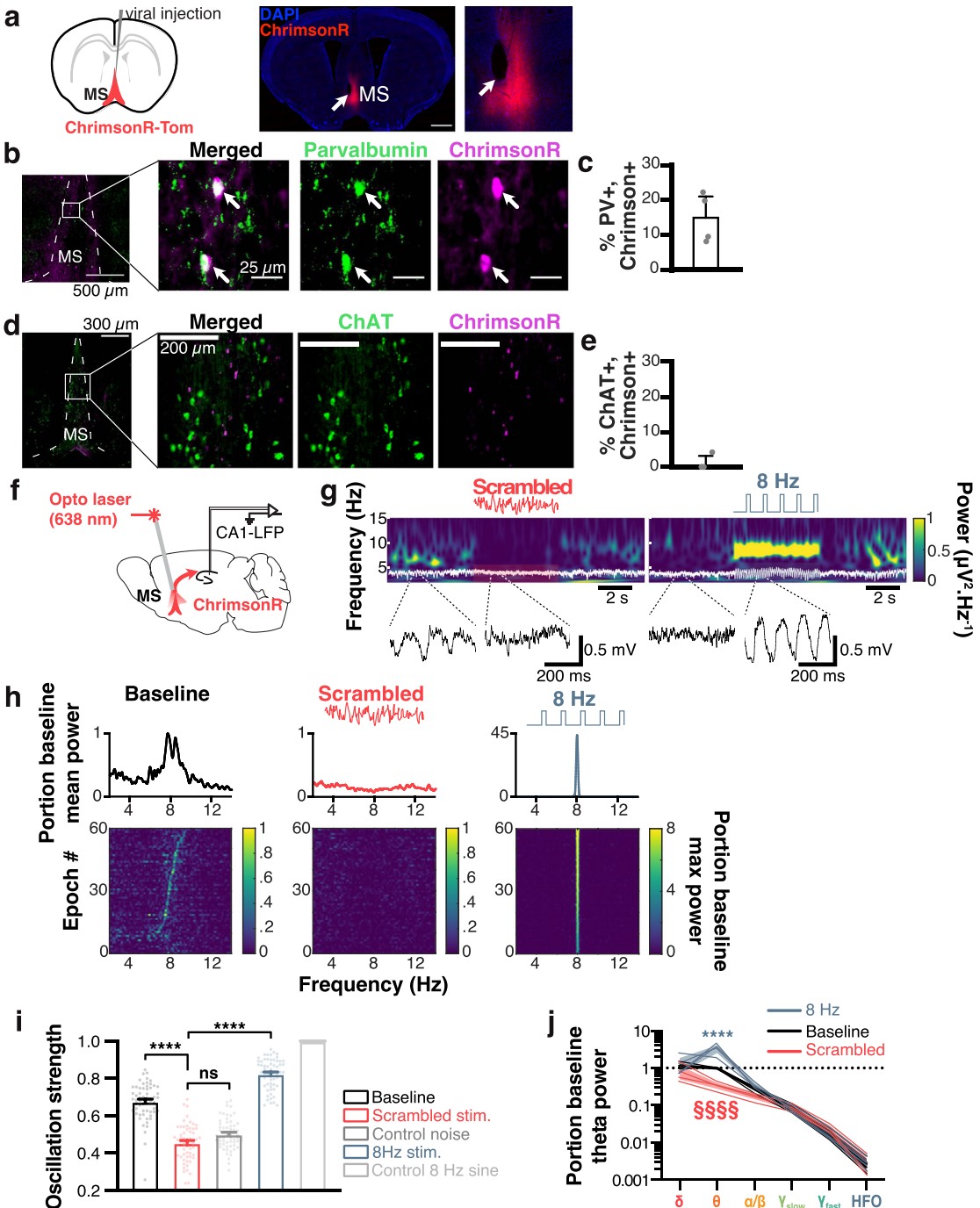

**Fig. 2 | Selective optogenetic pacing and ablation of theta oscillations. a** AAV-ChrimsonR expressing vector was injected in the MS (blue, DAPI; red, ChrimsonR), and a fiber optic was placed at the injection site (white arrow). Scale bar, 1 mm (MS, medial septum). **b** Immunohistological double staining for ChrimsonR (magenta) and parvalbumin (PV; green) neurons. **c** Quantification of ChrimsonR+/PV+ co-expressing neurons ($N = 4$ mice). **d** Immunohistological double staining for ChrimsonR (magenta) and cholinergic (ChAT+; green) neurons. **e** Quantification of ChrimsonR+/ChAT+ co-expressing neurons ($N = 4$ mice). **f** Recording configuration, with fiber optic in the MS and electrode in CA1. **g** Wavelet spectrograms before, during, and after scrambled (left, in red) and 8 Hz (right, in blue) stimulation patterns, with associated unfiltered LFP (white overlay, and enlarged, black traces at the bottom; dark blue, low power; bright yellow, high power). **h** Average Fourier power spectra for 60 × 5 s stimulation epochs (top), and sorted power spectra for each stimulation epoch (bottom; dark blue, low power; bright yellow, high power) for baseline (left, in black), scrambled (center, in red), and 8 Hz (right, in blue) stimulations in a representative mouse. **i** Oscillation strength for baseline vs. scrambled or 8 Hz optogenetic stimulations. Values for white noise (medium gray) and control 8 Hz sine waves are used for comparison (1ANOVA, $F_S = 251.5$, $p \leq 0.0001$, $n = 59$ independent epochs; Kruskal–Wallis multiple comparison tests; ****$p \leq 0.0001$). **j** Group average portion of baseline power for each stimulation pattern and every frequency band (log scale). Frequency bands: $\delta$, 1–4 Hz; $\theta$, 4–12 Hz; $\alpha/\beta$, 12–30 Hz; $\gamma_{slow}$, 30–60 Hz; $\gamma_{fast}$, 60–120 Hz; HFO (high-frequency oscillations), 120–250 Hz (2ANOVA, $F_{(10,24)} = 8.982$, $p \leq 0.0001$ for interaction between treatment and frequency band, $N = 3$ mice; Tukey's multiple comparisons test). **** or §§§§, $p \leq 0.0001$. All bar and line plots represent the mean ± SEM of at least two independent experiments.

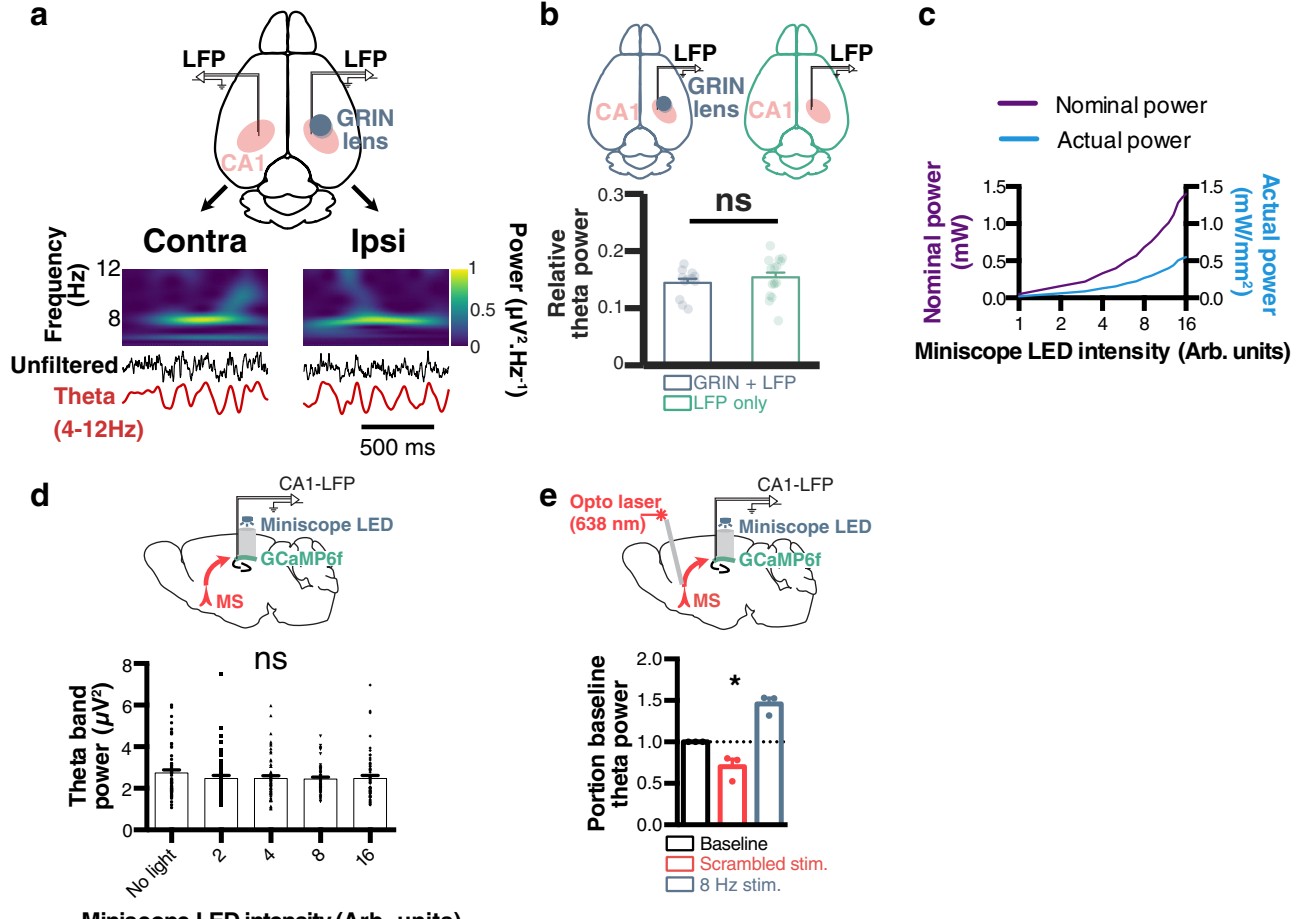

**Fig. 3 | GRIN lens implant does not affect theta oscillations and enables combined calcium imaging and optogenetics. a** Diagram showing recording configuration (horizontal plane), with local field potential (LFP) electrodes in CA1 of both hemispheres, and a GRIN lens only in the right hemisphere (top). For example, theta oscillations were recorded in both hemispheres using wavelet spectrograms (bottom; dark blue, low power; bright yellow, high power), raw (black), and theta filtered (red) traces. **b** Relative theta power measured in the right hemisphere of mice implanted with electrodes and GRIN lenses (blue; $N = 3$ mice) versus electrodes only (green; $N = 4$ mice; $t$-test, $t_{54} = 1.06$, $p = 0.2936$). **c** The relationship between miniscope LED input value and measured nominal power (purple) as well as actual light power corrected for the surface of a GRIN lens (blue). **d** Corresponding theta power from LFP electrode attached to GRIN lens when using different miniscope LED emitting light power values in the absence of optogenetic stimulations (1ANOVA, $F_4 = 0.7729$, $p = 0.5435$; $n = 60$ independent epochs). **e** Effects of MS optogenetic stimulations on theta oscillations during miniscope blue light emission through the lens (8 Arb. units LED power, corresponding to ~0.3 mW/mm²; Friedman test, $\chi^2 = 6.000$; $p = 0.0278$; $N = 3$ mice). *$p \leq 0.05$; ns not significant. All bar plots represent the mean ± SEM of at least two independent experiments.

($R^2 = 0.6$, $p \leq 0.0001$ for rest periods; $R^2 = 0.632$, $p \leq 0.0001$ for running periods; $n = 1849$ cells, $N = 5$ mice; Fig. 4c). Overall, -6.42 ± 0.52% of total cells were significantly modulated by scrambled optogenetic stimulations (Fig. 4d). Among those modulated cells, 50.56 ± 6.38% were inhibited while 49.43 ± 6.38% were excited ($n = 1849$ cells, $N = 5$ mice; Fig. 4e).

We next analyzed the effects of optogenetic stimulation on the spatial tuning of hippocampal neurons as mice freely explored the open field. To this end, we computed activity rate maps by using either epoch within or outside of stimulation periods (for the baseline condition, we included epochs that followed the same 5 s ON, 5 s OFF pattern used for actual stimulation; Fig. 4f). Stability was then computed as the correlation between rate maps for baseline and stimulation epochs. In spite of the aforementioned changes in overall activity, rate maps displayed no change of spatial stability for either scrambled or 8 Hz stimulations (Kruskal–Wallis $H_3 = 3.5$, $p = 0.1773$; Fig. 4g).

### MS optogenetic stimulation alters behavior but not spatio-temporal codes

To assess the effects of theta disruption on temporal and spatial codes, we monitored CA1 pyramidal neuron activity on the 3-tone linear track

(Fig. 5a). Here, we leverage one of the main advantages of calcium imaging, which is the ability to register recorded cells over several days while performing scrambled or 8 Hz stimulations on selected days (Figs. 5b, c, bottom panel). We focused our analysis on pairs of days with the same amount of time (48 h) between testing (Fig. 5c, top panel). For each condition, we assessed the portion of total cells significantly encoding one or several variables, and found no effect of either 8 Hz or scrambled stimulation on spatial and temporal encoding (RM-ANOVA; $F_2 = 0.807$, $p = 0.453$ for the main effect of stimulations; $F_6 = 1.283$, $p = 0.285$ for the interaction between stimulation and encoded variable, $n = 5$ mice; Fig. 5d).

While the portion of cells did not change under stimulation conditions, we assessed the effect of MS stimulation on place-, time-, and distance-modulated cells' tuning curves stability. To this end, we tracked neurons across days (Fig. 5c; see Methods) and computed the stability of place and time fields as the pairwise correlation between fields across 48 h. Because CA1 is known to display prominent remapping over days[58], we also used a pair of days with no stimulation to compute a baseline stability score to be used as a reference (Fig. 5e–g). We found no change in stability during MS stimulation for place-modulated cells (1ANOVA, $F_2 = 1.907$, $p = 0.1511$; $n = 205$ cell pairs

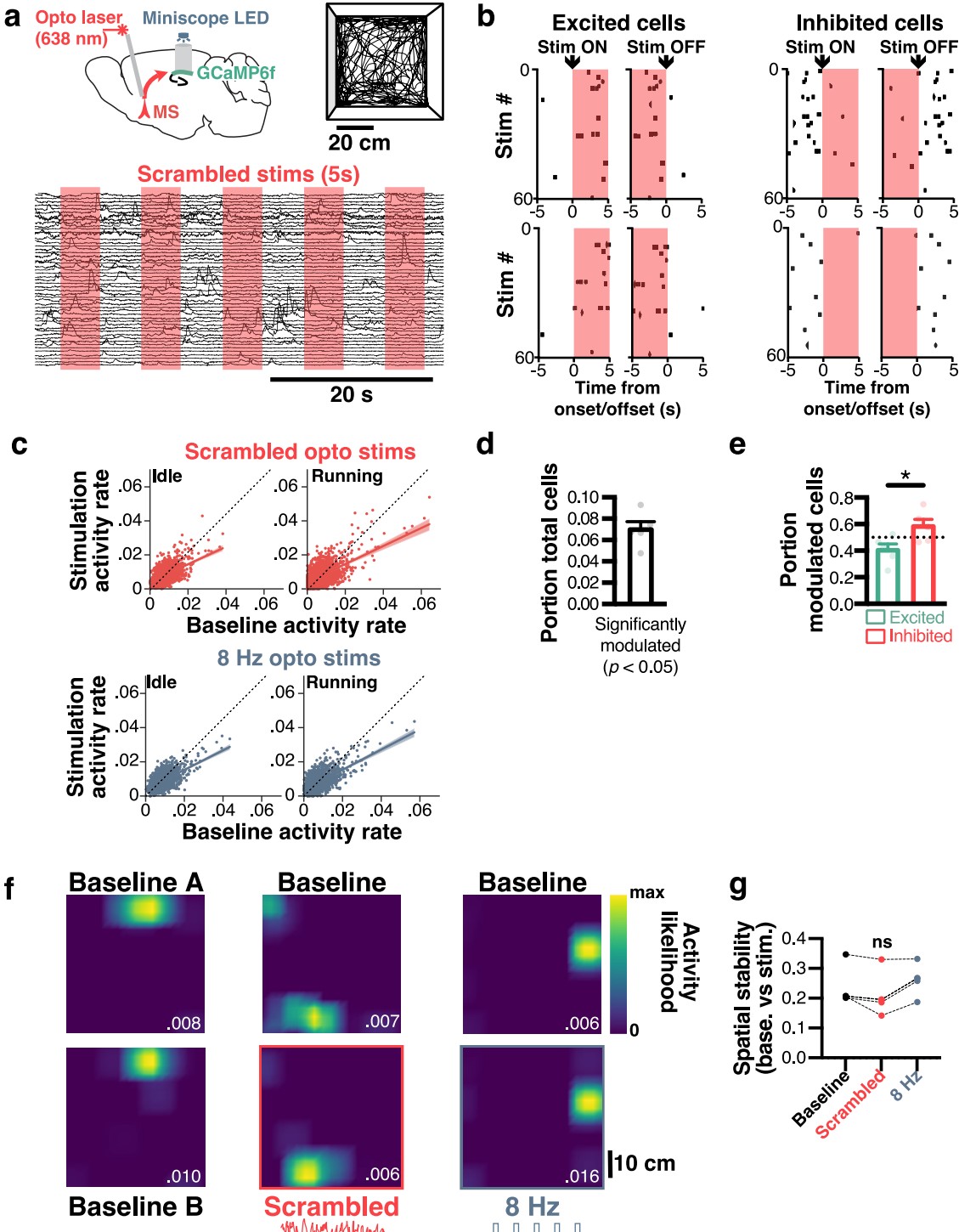

**Fig. 4 | MS optogenetic stimulation modulates the activity of a subpopulation of CA1 pyramidal cells. a** Recording CA1 neurons during MS optogenetic stimulation (5 s ON, 5 s OFF) while simultaneously imaging CA1 neurons in mice freely exploring an open field. **b** Peristimulus time raster plot of binarized activity for pairs of excited (left) or inhibited (right) neurons. **c** Scatter plot for all pooled neurons ($n = 1849$ cells, $N = 5$ mice) of activity probability during baseline vs stimulation (red, top: scrambled; blue, bottom: 8 Hz), for both idle (left) and running (right) periods. **d** Portion of total cells significantly modulated by stimulations ($N = 5$ mice). **e** Portion of modulated cells that are excited (green) or inhibited (red; two-tailed unpaired $t$-test, $t_8 = 2.511$, $p = 0.0363$, $N = 5$ mice). **f** Spatially tuning of example neurons before (top panels) or during (bottom panels) stimulations. For the baseline condition, comparable 5 s epochs were used to sample spatial tuning (baseline A vs. B). Dark blue, low activity likelihood; bright yellow, high activity likelihood. The top number indicated peak activity likelihood for each neuron. **g** Average stability of spatial tuning between stimulated and unstimulated epochs during baseline scrambled stimulation, and 8 Hz stimulation sessions (Kruskal–Wallis test, $F_3 = 3.5$, $p = 0.1773$, $N = 5$ mice). ns not significant. All bar plots represent the mean ± SEM of at least two independent experiments. Bands of line plots in **c** represent 95% confidence intervals.

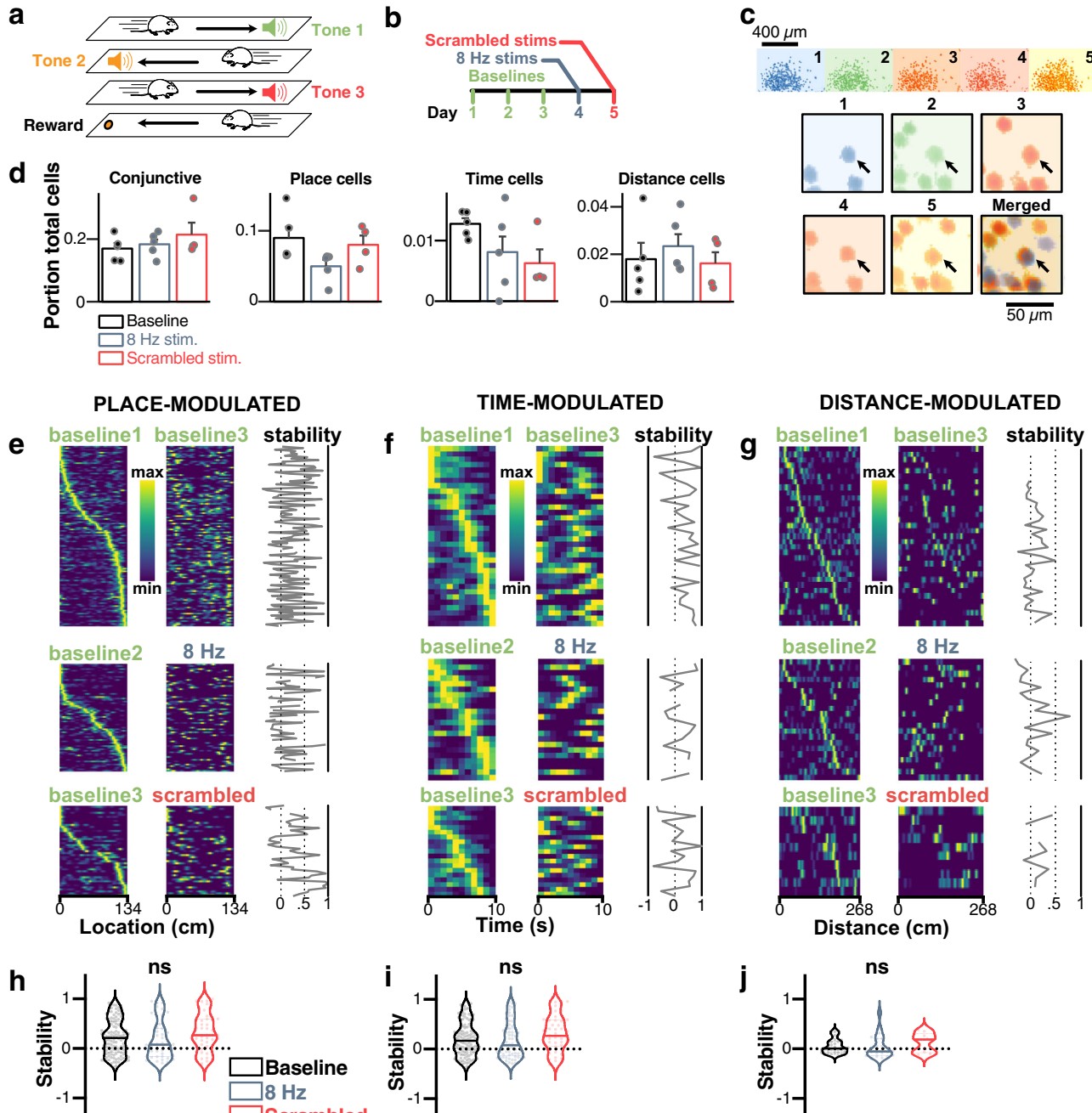

**Fig. 5 | MS optogenetic stimulation does not disrupt the stability of spatio-temporal codes. a** We recorded mice on the three-tone linear track to identify time-, place-, and distance-modulated cells (see Fig. 1 and Methods). **b** Experimental timeline. **c** Spatial footprints of neurons recorded over days and comparison scheme used to assess stability (top). Example neuron (black arrow) tracked over the five experimental days (bottom). **d** Portion of total cells encoding for one or several (conjunctive) variables before and during 8 Hz and frequency scrambling optogenetic stimulations (N = 5 mice). **e** Sorted place fields for identified place cells across day pairs. The first day is used as a baseline and mice undergo 8 Hz, scrambled, or no stimulations (control treatment) during the other day. Stability is computed as the pairwise correlation of fields between the two-day pairs. **f** Same for time-modulated cells. **g** Same for distance-modulated cells. The color scale in **e**–**g**: dark blue, low activity; bright yellow, high activity. **h** Corresponding average stability for place-modulated cells (1ANOVA, $F_2 = 1.907$, $p = 0.1511$; n = 205 cell pairs pooled from N = 5 independent mice). **i** Same for time-modulated cells (1ANOVA, $F_2 = 2.201$, $p = 0.1130$; n = 227 cell pairs pooled from N = 5 independent mice). **j** Same for distance modulated cells (1ANOVA, $F_2 = 0.6962$, $p = 0.5024$; n = 64 cell pairs pooled from N = 5 independent mice). ns not significant. All bar plots represent mean ± SEM of at least three independent experiments.

pooled from N = 5 independent mice; Fig. 5h), time-modulated cells (1ANOVA, $F_2 = 2.201$, $p = 0.113$; n = 227 cell pairs pooled from N = 5 independent mice; Fig. 5i), and distance-modulated cells (1ANOVA, $F_2 = 0.6962$, $p = 0.5024$; n = 64 cell pairs pooled from N = 5 independent mice; Fig. 5j). We also extended the same analysis to conjunctive neurons (i.e. neurons that can encode more than one variable; Supplementary Fig. 5a–f) and found no effects of optogenetic stimulations

on the stability of conjunctive spatial (1ANOVA, $F_2 = 3.731$, $p = 0.0661$; N = 4 independent mice; Supplementary Fig. 5g), temporal (1ANOVA, $F_2 = 1.993$, $p = 0.8228$; N = 4 independent mice; Supplementary Fig. 5h), and distance cells (1ANOVA, $F_2 = 0.469$, $p = 0.6400$; N = 4 independent mice; Supplementary Fig. 5i).

We next assessed the quality of spatiotemporal codes using a naive Bayesian classifier to decode location (Fig. 6a, b), time (Fig. 6c,

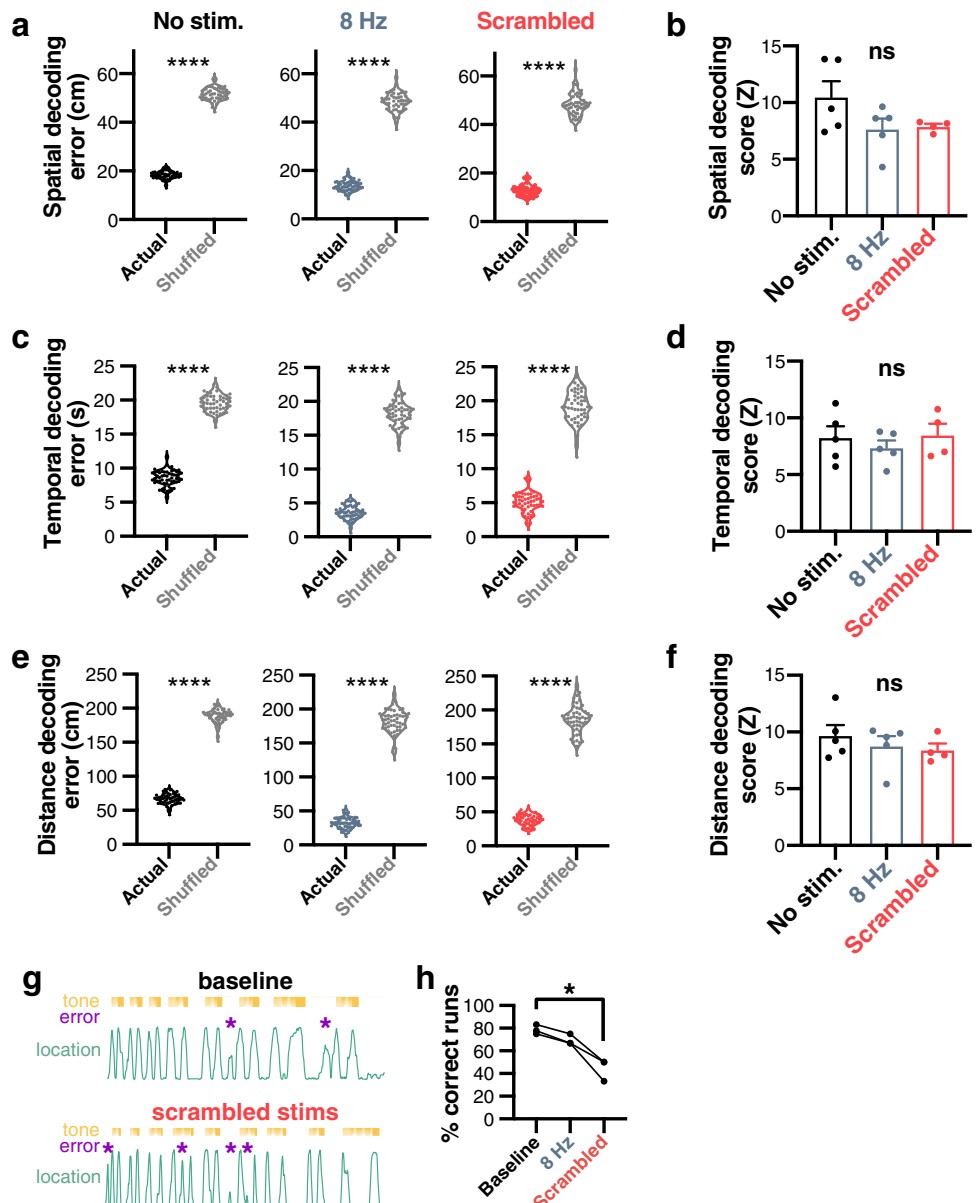

**Fig. 6 | MS optogenetic stimulation alters behavior but does not disrupt spatiotemporal codes.** Using a naive Bayesian classifier, location (**a, b**), the time elapsed (**c, d**), and distance traveled (**e, f**) were predicted from CA1 neuronal activity. **a** Decoding error (euclidean distance) for baseline, 8 Hz, and scrambled optogenetic stimulation conditions for location (2ANOVA, $F_5 = 2172$, $p \leq 0.0001$; $n = 50$ bootstrap samples from one representative mouse). **c** Same for time (2ANOVA, $F_5 = 1292$, $p \leq 0.0001$; $n = 50$ bootstrap samples from one representative mouse). **e** Same for distance (2ANOVA, $F_5 = 1964$, $p \leq 0.0001$; $n = 50$ bootstrap samples from one representative example mouse). **b** Decoding $z$-score (higher values indicate better decoding compared to shuffled surrogate) for location (1ANOVA, $F_2 = 2.332$, $p = 0.1432$; $N = 5$ mice). **d** Same for time (1ANOVA, $F_2 = 0.4561$, $p = 0.6452$; $N = 5$ mice). **f** Same for distance (1ANOVA, $F_2 = 0.6102$, $p = 0.5606$; $N = 5$ mice). **g** Example scoring of errors (purple stars) in the tone-cued linear track task. **h** Percent of correct trials during baseline, 8 Hz, and scrambled stimulations (Friedman test, $\chi^2 = 6.000$, $p = 0.0278$; $N = 3$ mice). \*\*\*\*$p \leq 0.0001$; ns not significant. All bar plots represent the mean ± SEM of at least two independent experiments.

d), and distance traveled (Fig. 6e, f), using random bootstrap samples ($n = 50$ bootstrap samples, $n = 160$ cells per sample). Decoded errors were systematically lower than shuffled surrogates, including during 8 Hz or scrambled stimulation for location (2ANOVA, $F_5 = 2172$, $p \leq 0.0001$; $n = 50$ bootstrap samples from one representative mouse; Fig. 6a), time (2ANOVA, $F_5 = 1292$, $p \leq 0.0001$; $n = 50$ bootstrap samples from one representative mouse; Fig. 6c), and distance (2ANOVA, $F_5 = 1964$, $p \leq 0.0001$; $n = 50$ bootstrap samples from one representative mouse; Fig. 6e) indicating that spatiotemporal codes were preserved during stimulation. To estimate the inter-individual significance of spatiotemporal codes, we $z$-scored decoding error by using both the actual and shuffled results (see Methods) for a given day, for every mouse (Fig. 6b, d, f).

MS optogenetic control did not significantly alter the encoding of location (1ANOVA, $F_{2,11} = 2.2332$, $p = 0.1432$; $N = 5$ mice; effect size $\eta^2 = 0.29$; Fig. 6b), time (1ANOVA, $F_{2,11} = 0.4561$, $p = 0.6452$; $N = 5$ mice; effect size $\eta^2 = 0.07$; Fig. 6d), or distance (1ANOVA, $F_{2,11} = 0.6102$, $p = 0.5606$; $N = 5$ mice; effect size $\eta^2 = 0.09$; Fig. 6f).

In the behavioral paradigm used to analyze temporal modulation of neuronal activity, mice are water-scheduled and trained to collect rewards. Based on this assumption, we quantified the number of returns to an empty reward site as errors, and computed the percentage of correct trials to total trials as a proxy for performance (Fig. 6g). In these conditions, we found a significant effect of optogenetic stimulation on performance across days (Friedman test $\chi^2 = 6.000$, $p = 0.0278$) and in particular a significant difference in

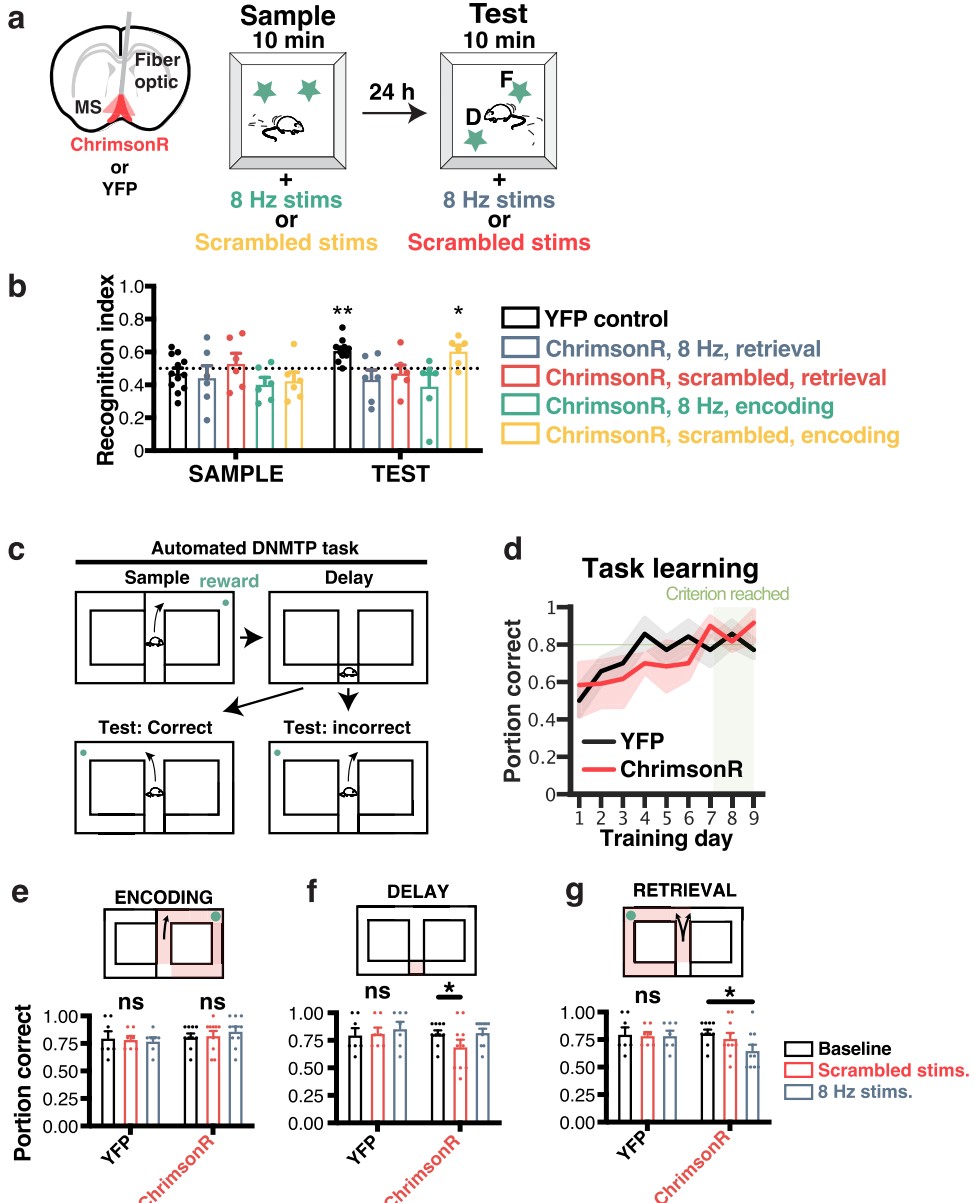

**Fig. 7 | MS optogenetic stimulations disrupt maintenance and retrieval, but not encoding of episodic and working memory. a** Mice were implanted with fiber optics in the MS after transfecting ChrimsonR (top). They were then subjected to the novel object place recognition task (bottom, see Methods). **b** In this task, both scrambled (red), and 8 Hz (blue) stimulations during retrieval, as well as 8 Hz (green) but not scrambled (yellow) stimulations during encoding disrupted memory performance (2ANOVA, $F_{4,31} = 3.283$ for the main effect of treatment, $p = 0.0097$; effect size for the main effect of group, $\eta^2_p = 0.306$; N = 12 mice). **c** To further examine the effect of MS stimulation on memory encoding, maintenance, and retrieval, mice were trained in a delayed non-match to sample (DNMTS) task in an automated T-maze. **d** Mice transfected with ChrimsonR (red) or YFP (black) were trained (in the absence of stimulation) to choose the correct, non-matching arm until the performance exceeded a criterion of 0.8 portions of correct choices

per day, for at least two consecutive days (green band; RM-ANOVA, $F_8 = 8.738$, $p \leq 0.0001$ for the main effect of training days; pairwise Tukey multiple comparison tests between groups, $p = 0.420$; N = 17 mice). **e–g** Performance during stimulation at different phases of the task for mice injected with ChrimonR (top) or YFP controls (bottom). Red shading indicates the stimulated regions of the maze. **e** Average daily performance when performing MS stimulations during encoding only (RM-ANOVA, $F_{11} = 2.197$, $p = 0.547$; N = 17 mice). **f** Average daily performance when performing MS stimulations during the 10 s delay period only (RM-ANOVA, $F_{11} = 3.483$, $p = 0.0495$; effect size for the main effect of treatment, $\eta^2_p = 0.109$; N = 17 mice). **g** Average daily performance when performing MS stimulations during retrieval only (RM-ANOVA, $F_{11} = 3.265$, $p = 0.050$; N = 17 mice). All bar plots and line plots represent the mean ± SEM of at least three independent experiments.

performance between during baseline (78.70 ± 2.45%) and scrambled stimulation (44.44 ± 5.56%; multiple comparisons, $p = 0.0429$; N = 3 mice; Fig. 6h). Only mice with a minimum of 12 runs were included in this analysis. Because of the limitations of this task (low cognitive load and a low number of mice tested), we next set out to assess the effect of MS optogenetic stimulation in standardized memory tasks.

## Disruption of theta signals impairs spatial recognition and working memory retrieval

To test the role of theta signals in spatial memory, we used a dedicated group of mice injected with ChrimsonR and implanted with a fiber optic in the MS (Fig. 7a, left panel). Mice were subjected to a novel place object recognition (NPOR) task (Fig. 7a, right panel). To assess the role of theta oscillations in memory encoding and retrieval, we

performed optogenetic stimulations specifically during the sample and test phases and computed the recognition index (RI; see Methods). When stimulating during retrieval, memory performance was significantly reduced for both scrambled (0.472 ± 0.048 RI, $n = 6$ mice) and 8 Hz stimulations (0.435 ± 0.057 RI, $n = 6$ mice) groups, compared to YFP controls that displayed a significant increase in object exploration during testing (0.61 ± 0.018 RI; 2ANOVA, $F_{4,31} = 3.283$ for the main effect of treatment, $p = 0.0097$; effect size for the main effect of group, $\eta^2_p = 0.306$; $N = 12$ mice; Fig. 7b). On the other hand, scrambled stimulations during encoding did not impair memory at test time (0.60 ± 0.034 RI, $p = 0.0157$, $n = 6$ mice) but 8 Hz stimulation during encoding lowered memory performance to chance levels (0.39 ± 0.074 RI, $p = 0.8740$, $n = 6$ mice).

To examine the effect of optogenetic control of the MS on specific phases of working memory function (encoding, maintenance, and retrieval), mice were transfected with ChrimsonR and implanted with fiber optics in the MS and trained in a delayed non-match to sample (DNMTS) task. In the sample phase, mice were forced to run to a randomly designated arm to collect a reward. After a delay (10 s), they could either run in the opposite arm (correct choice) to receive another reward or run in the same, unrewarded arm (incorrect arm; Fig. 7c). The advantage of this task is to allow for repetitive testing, specific isolation of task phases (training, delay, testing) and within-subject controls. Mice were trained in this task with no stimulation until a criterion performance of 0.8 (a portion of correct trials) was reached for at least two days. Both ChrimsonR and YFP control mice displayed significant improvement over time (RM-ANOVA, $F_8 = 8.738$, $p \leq 0.0001$ for the main effect of training days). Importantly, we found no learning rate differences between the two groups (pairwise Tukey; $p = 0.420$; Fig. 7d). Once mice had learned the rule associated with the DNMTS task, we assessed performance while delivering either scrambled (0.792 ± 0.045) or 8 Hz (0.850 ± 0.036) optogenetic stimulation in the encoding phase (forced choice) only, and did not observe any difference in performance compared to baseline (0.825 ± 0.030, RM-ANOVA, $F_{11} = 2.197$, $p = 0.547$, $N = 17$; Fig. 7e). When stimulated only during the delay period, only scrambled stimulations significantly decreased memory performance (0.733 ± 0.057) compared to baseline (0.825 ± 0.030, RM-ANOVA, $F_{11} = 3.483$, $p = 0.0495$; effect size for the main effect of treatment, $\eta^2_p = 0.109$; Fig. 7f). In contrast, when stimulated during retrieval, mice stimulated with 8 Hz displayed significantly reduced memory performance (0.675 ± 0.049) compared to baseline (0.825 ± 0.030, RM-ANOVA, $F_{11} = 3.265$, $p = 0.050$; Fig. 7g). In contrast, YFP control mice were not affected by stimulations during encoding (RM-ANOVA, $F_2 = 0.1314$, $p = 0.8781$), the delay period (RM-ANOVA, $F_2 = 0.2020$, $p = 0.8197$), or retrieval (RM-ANOVA, $F_2 = 0.0454$, $p = 0.9557$; effect size for the main effect of treatment, $\eta^2_p = 0.196$) of working memory.

### Optogenetic control of MS neurons does not alter locomotion

Importantly, it was previously reported that pacing theta oscillations could decrease locomotor speed and its variability[10], which could explain at least in part the effects of optogenetic stimulations on working and episodic memory. To thoroughly assess the specificity of MS optogenetic stimulation on memory, we performed additional experiments to rule out the direct effects of the optogenetic stimulations on locomotor velocity. A subset of mice injected with ChrimsonR and implanted with fiber optics in the MS as well as LFP electrodes in CA1 were allowed to explore an open field freely while being subjected to 5 s ON, 5 s OFF optogenetic stimulations (Supplementary Fig. 6a). 8 Hz stimulation led to consistent pacing of hippocampal oscillations to that frequency (Supplementary Fig. 6b). Those stimulations were not associated with any apparent change in locomotor behavior (Supplementary Fig. 6c), including mean speed (unpaired, two-tailed $t$-test, $t_{116} = 0.4140$, $p = 0.6796$; Supplementary Fig. 6d) and speed coefficient of variation (CV; unpaired, two-tailed $t$-test, $t_{116} = 0.8296$,

$p = 0.4095$, $n = 59$ stimulation epochs; Supplementary Fig. 6e). Similarly, scrambled stimulation consistently led to abolished theta oscillations (Supplementary Fig. 6f) but no apparent changes in locomotor behavior (Supplementary Fig. 6g), including mean speed (unpaired, two-tailed $t$-test, $t_{118} = 0.2268$, $p = 0.8210$; $n = 60$ stimulation epochs; Supplementary Fig. 6h) and speed CV (unpaired, two-tailed $t$-test, $t_{118} = 1.838$, $p = 0.686$; $n = 60$ stimulation epochs; Supplementary Fig. 6i).

While natural theta frequency and locomotor speed are correlated[59], the exact direction of causation between these variables remains poorly understood. To answer this question, we performed optogenetic stimulation using a 5 s ON, 5 s OFF paradigm, and a random frequency was selected for each stimulation epoch (Supplementary Fig. 6j). These stimulation frequencies covered the entirety of the theta band spectrum (Supplementary Fig. 6k). As expected, we found that natural theta oscillations frequencies are directly correlated to running speed for one example mouse ($R^2 = 0.1114$, $p \leq 0.0001$; Supplementary Fig. 6l). Importantly, when theta oscillation frequency results from MS optogenetic control, the correlation drops below chance level ($R^2 = 0.0006779$, $p = 0.6244$; Supplementary Fig. 6m) suggesting that locomotion dictates theta frequency, but not the opposite. We systematically replicated these results across mice and observed a drop in the correlation between theta frequency and locomotor speed under optogenetic stimulation control (paired $t$-test, $t_3 = 3.922$, $p = 0.0295$; $N = 4$ mice; Supplementary Fig. 6n). Altogether, these results support that our optogenetic stimulation does not alter locomotor behavior, which is highly relevant for the next behavioral assays.

### Optogenetic pacing of MS neurons leads to non-physiological theta synchrony

Surprisingly, 8 Hz optogenetic stimulations that are associated with the pacing of theta oscillations led to decreased performance when applied during either encoding or retrieval of episodic memory in the NPOR task, as well as spatial working memory retrieval in the DNMTS task. To further understand the effects of pacing theta on hippocampal physiology, we analyzed phase locking of theta oscillation before and during 8 Hz optogenetic stimulations (Supplementary Fig. 7a) and found that such stimulation led to increased phase synchrony across stimulation epochs (Supplementary Fig. 7b). Furthermore, we analyzed the cross-correlation of theta oscillations across dorsal CA1 (~1 mm distance between electrodes along the septotemporal axis (Supplementary Fig. 7c) and found that pacing theta rhythms at 8 Hz led to an increased cross correlation between those two electrodes (Supplementary Fig. 7d).

## Discussion

Within the septohippocampal system, the exact causal relationships between (1) MS activity, (2) hippocampal oscillations, (3) hippocampal neuron activity, and (4) behavior, including memory, remain an active area of research. In particular, whether and how the MS supports encoding of place and time in the hippocampus, as well as its specific contribution to memory function, remain unclear. Here, we leveraged more recent techniques that allowed us to record >1000 neurons while performing optogenetic stimulation of MS neurons with a sub-second resolution to control theta rhythms and assess their role in hippocampal physiology and memory. Importantly, using an excitatory opsin and either scrambled or 8 Hz stimulations, we were able to consistently and robustly abolish or pace hippocampal theta, respectively. Compared to MS inhibition using either an inhibitory opsin or pharmacological compounds (e.g. muscimol), our approach enables a within-subject comparison of two opposed states (paced vs abolished theta) while maintaining activity levels in MS-PV cells. We combined a tone-cued linear track task that, together with information-theoretic approaches, allowed us to disentangle spatial and temporal hippocampal codes. This method alleviates the need for arbitrary

thresholds, enabling a standardized approach to calcium imaging analysis[52]. While a large portion of CA1 pyramidal neurons expressed a mixture of spatiotemporal codes, we focused our analyses on neurons tuned specifically to place, time, or distance traveled.

While spatial coding has been extensively investigated in CA1, temporal and distance codes have recently gained more interest. Temporal[28,35–37] and distance[37] codes have been extracted by clamping visuospatial cues or extracted analytically using generalized linear models in virtual reality paradigms[26,27,38]. Here, we propose an approach to disentangle spatial, temporal, and distance codes using an information theoretic approach which, together with our cued-alternation task, enables the analysis of real-world spatiotemporal and multiplexed codes in freely moving animals. A large amount of CA1 pyramidal neurons encode multiplexed information of location, distance, and time as previously reported[26,27,38] in addition to self-motion signals such as acceleration, speed, and orientation[60].

We found that frequency scrambling and 8 Hz optogenetic stimulations drastically abolished or paced theta rhythms, respectively, and led to decreased overall activity in a subpopulation of CA1 pyramidal cells while not causing significant changes in place cell activity, similar to previous reports using either pharmacological inhibition[36,46,47], or optogenetic[9,48] pacing of MS or septal inputs to the hippocampus. As MS neurons are known to be the primary driver of hippocampal oscillations, we expected that MS stimulation would be associated with disrupted time or distance cell activity in conditions of reduced theta rhythms, but no changes occurred when theta was abolished or paced. Additionally, the pacing of hippocampal oscillations to 8 Hz did not lead to changes in the quality of spatial codes as reported previously[9] and did not alter temporal codes as observed in our behavioral paradigm.

Although it was reported that time (but not place) cells could depend directly on the medial entorhinal cortex (MEC) inputs[61], recent experimental evidence suggests that MEC lesions do not lead to any alterations in hippocampal time cell physiology[62]. Notably, a more recent investigation found that optogenetic pacing of the MS did not disrupt the spatial codes of grid cells in the entorhinal cortex[63], further suggesting that MS activity is not directly involved in spatial codes within the hippocampal formation. Together with our results, this indicates that temporal codes could either be the result of MS-dependent computations within the entorhinal cortex[63] (1) or be generated intrinsically within the hippocampus itself (2).

While early studies found that the MS played an essential role in maintaining time cells and supporting working memory[36], recent experimental evidence shows that the exact contribution of time cells to working memory performance could be less than previously thought[62]. Since MS-PV neurons likely maintain significant activity levels during frequency scrambling optogenetic stimulation (in contrast to inhibition approaches), our results strongly support the role of precisely timed septal activity in supporting working memory, as hypothesized previously[22]. Although we observed reduced memory performance when using optogenetic stimulation during retrieval, stimulation during the encoding phase of the DNMTS tasks was not associated with reduced memory in the DNMTS task. Pacing theta oscillations at 8 Hz during either the encoding or retrieval phase of the NOPR task led to impaired memory retrieval. Our electrophysiological analyses further reveal that such stimulation caused distant regions of dorsal CA1 to be entrained at the same phase, with non-physiological phase locking. While we did not directly investigate the causal relationship between phase changes and memory performance, spike timing, and phase precession were also found to be associated with altered working memory function in spite of leaving place coding intact[64]. Moreover, phase locking of pyramidal neurons to theta oscillations has previously been shown to be a predictor of memory performance[65].

Notably, one limitation of our approach to establishing temporal and distance tuning properties is that it requires four times more sampling than a regular linear track which increases the chances of photobleaching when adding stimulation conditions to the experimental timeline. While we found that spatial codes remained unaffected by optogenetic stimulations within the same session, more recent imaging hardware that includes sensors with increased sensitivity could help prevent photobleaching and allow for longer recording sessions, thus sampling of temporal and distance cells along with stimulations. We also found that the pool of neurons that significantly and specifically encode only one variable is small in comparison to the number of conjunctive neurons, making sampling requirements for these highly specific neurons much higher.

We provide experimental evidence that manipulation of MS does not alter the locomotor speed and that, conversely, locomotor speed does dictate theta frequency. Even though the place and time codes were preserved during our MS stimulations, a portion (~6%) of cells were directly modulated and could account for memory impairments observed in our behavioral assays. PV interneurons in the hippocampus have previously been reported to be part of a microcircuit essential in regulating memory consolidation[66,67], and our optogenetic manipulations could be associated with the silencing of a portion of these cells. Moreover, although we did not observe changes in frequency bands other than theta, we cannot exclude that CA1 pyramidal neuron spike timing could be drastically altered when scrambling theta oscillations, while 8 Hz stimulation was shown to not result in increased hippocampal activity[18], which could explain the differential effects of those stimulation patterns on working memory performance. The memory impairments we observed were likely non-cholinergic: firstly, in our immunohistological experiments, we found virtually no expression of ChrimsonR in ChAT neurons of the MS. Secondly, our optogenetic stimulations were not associated with changes in hippocampal ripples, while previous reports indicate that stimulation of ChAT neurons is associated with reduced ripple frequency[45,57]. ChrimsonR transfection of MS VGLUT2 neurons is also unlikely since activation of these neurons is associated with a direct increase in locomotor activity[68], which we did not observe. Although the precise temporal arrangement of CA1 pyramidal spikes could explain, at least partially, the effects of MS stimulation on memory, alternative mechanisms should be taken into consideration. Notably, in addition to the hippocampus and the entorhinal cortex, MS PV neurons project to the retrosplenial cortex[3], and could be responsible for some of the memory impairments observed here.

In summary, using calcium imaging, optogenetics, and electrophysiology, we found that theta rhythms can be paced or abolished using stimulation of the MS. Such stimulation impaired episodic as well as working memory retrieval. These effects were non-cholinergic and did not disrupt hippocampal ripple activity. Finally, while a small fraction of hippocampal neurons responded directly to optogenetic stimulation of the MS, place, time, and distance cells were not disrupted by manipulations of theta oscillations. Together these results suggest that while MS input to the hippocampus plays an essential role in memory, multiplexed codes in CA1 pyramidal neurons might not be the direct substrate for such functions.

## Methods

### Animals

All procedures were approved by the McGill University Animal Care Committee and the Canadian Council on Animal Care (protocol 2015-7650). A total of $n = 41$ male ($n = 20$) and female ($n = 21$) 8–16 weeks old, B6;129P2 PV-Cre mice (Jackson Laboratory, RRID:IMSR_JAX:017320) were used in this study. $n = 5$ mice were used for combining optogenetics with calcium imaging; $n = 3$ mice were implanted for calcium imaging, optogenetics, and electrophysiological controls; $n = 4$ mice were used in electrophysiological studies; $n = 29$ mice were transfected

and implanted for behavioral assays. Mice were housed individually on a 12-h light/dark cycle at 22 °C and 40% humidity with food and water ad libitum. All experiments were carried out during the light portion of the light/dark cycle.

## Adeno-associated viral vectors

Adeno-associated virus AAV5.CamKII.GCaMP6f.WPRE.SV40 (Addgene # 100834, obtained from the University of Pennsylvania Vector Core) was used in all calcium imaging experiments. Adeno-associated virus (AAV) of serotype *dj* (hybrid capsid created from eight different AAV serotypes) AAVdj-hSyn-ChrimsonR-tdTomato were obtained from the Vector Core Facility at Oregon Health and Science University in Portland, Oregon. Although hSyn is a housekeeping gene, we did not observe transfection in cholinergic cells (see "Results" and Fig. 2b–e). An eYFP construct without the ChrimsonR sequence was used as a control (termed YFP control in this manuscript).

## Surgical procedures

Mice were anesthetized with isoflurane (5% induction, 0.5–1 % maintenance) and placed in a stereotaxic frame (David Kopf Instruments). Body temperature was maintained with a heating pad, eyes were hydrated with gel (Optixcare), and Carprofen (10 ml/kg) was administered subcutaneously. The skull was completely cleared of all connective tissue and small craniotomies were performed using a dental drill for subsequent injection or implant.

**Viral injections**. All viral injections were performed using a glass pipette connected to a Nanoject III (Drummond) injector. 500 nl of AAVdj-hSyn-ChrimsonR-tdTomato (or eYFP control) was delivered into the MS at a rate of 1 nL/s, at the following coordinates based on reference mouse stereotaxic atlas[69] (distance from Bregma in mm): anteroposterior (AP) 0.85, mediolateral (ML) 0, dorsoventral (DV) −4.50 using a 5° angle in the ML plane. After surgery, animals were monitored until recovery.

**Fiber optic implant**. Two weeks post-injection, mice were anesthetized for implantation following the same surgical procedure. A 200 μm diameter fiber optics with ceramic ferrule (Thorlabs) was implanted at the same coordinates. Implants were cemented in place using C&B-Metabond® (Patterson dental). Black nail polish was applied over the dental cement to block light emission during optogenetic stimulation.

**Electrode implants**. An array of 7 tungsten microelectrodes (~1 MΩ impedance) was lowered in dorsal CA1 spanning through *stratum pyramidale* (pyr), *stratum radiatum* (rad), and *stratum lacunosum moleculare* (lm). Screws placed in the bone above the frontal cortex and cerebellum served as ground and reference, respectively. Following electrode, ground, and reference placement, dental cement was applied to secure the implant permanently to the skull.

**Implant for calcium imaging**. We injected the AAV5.CamKII.GCaMP6f virus (200 nL at 1 nl s⁻¹) in hippocampal CA1 using the following coordinates: anteroposterior (AP) − 1.86 mm from bregma, mediolateral (ML) 1.5 mm, dorsoventral (DV) 1.5 mm. Two weeks following the injection, mice were anesthetized with isoflurane and the skull was cleared. A <2 mm diameter craniotomy was performed in the skull above the injection site. An anchor screw was placed on the posterior plate above the cerebellum. The dura was removed, and the portion of the cortex above the injection site was aspirated using a vacuum pump until the corpus callosum was visible. These fiber bundles were then gently aspirated without applying pressure on the underlying hippocampus, and a 1.8 mm diameter gradient index (GRIN; Edmund Optics) lens was lowered at the following coordinates: AP − 1.86 mm from bregma, ML 1.5 mm, DV 1.2 mm. The GRIN lens was permanently

attached to the skull using C&B-Metabond (Patterson Dental), and Kwik-Sil (World Precision Instruments) silicone adhesive was placed on the GRIN to protect it. Four weeks later, the silicone cap was removed and CA1 was imaged using a miniscope mounted with an aluminum base plate while mice were under light anesthesia (<0.5% isoflurane) to allow the visualization of cell activity. When a satisfying field of view was found (large neuronal assembly, visible landmarks), the base plate was cemented above the GRIN lens, the miniscope was removed, and a plastic cap was placed on the base plate to protect the GRIN lens.

**Simultaneous calcium imaging and electrophysiological recordings**. To control for the effects of GRIN implants on hippocampal theta as well as cross-talk between miniscope and optogenetic excitation lights, we attached tungsten micro-electrodes to GRIN lenses. To this end, we placed GRIN lenses horizontally under a low-magnification microscope in a dust-free environment. A tungsten microelectrode was gently placed on the top edge of the GRIN lens using a micro-manipulator. We used the known diameter of the GRIN lens as a reference unit to estimate the desired protrusion of the electrode (~50 μm, further assessed digitally after taking a microphotograph of the preparation) in accordance with our planned implantation coordinates. Small amounts (~50 μL) of superglue were deposited on the top edge of the GRIN lens with the electrode in place and left to dry for ~10 min, before applying the next layer of glue. Five thin layers were used to maintain the electrode attached to the GRIN lens. After implantation of these GRIN-electrode assemblies using the protocol described above, the protruding wires were gently bent and hidden under a protective cap (1–1.5 cm high) retrieved from the pear of a manual suction pipette as a replacement for Kwik-Sil.

## In vivo behavioral procedures

**Habituation**. Mice were gently handled for ~5 min over the course of one week, with progressive habituation to the plugging procedure (fiber optic, miniscope, and electrophysiological pre-amplified tethers). Animals were then water-scheduled (2 h access per day).

**Miniscope recordings**. Miniscopes (V3) were assembled using open-source instructions (miniscope.org). Imaging data were acquired using a CMOS imaging sensor (Aptina, MT9V032) and multiplexed through a lightweight coaxial cable. Data was acquired using a data acquisition (DAQ) box connected via a USB host controller (Cypress, CYUSB3013). Animal behavior was recorded using a consumer-grade webcam (Logitech) mounted above the environment. Calcium and behavioral data were recorded using miniscope.org open, source DAQ custom acquisition software. The DAQ simultaneously acquired behavioral and cellular imaging streams at 30 Hz as uncompressed.avi files of 1000 frames for 15 min recording sessions, along with corresponding time-stamps to enable precise temporal alignment of behavioral and calcium imaging data.

**In vivo electrophysiological recordings**. Following 1 week of post-surgical recovery and one week of habituation to the tethering setup, LFP from implanted mice was recorded. All recorded signals from implanted electrodes were amplified by the tether pre-amplifier before being digitized at 22 kHz using a digital recording system (Neuralynx, USA). Recordings of each channel signal were saved along with video recordings and TTL signals from the Laser Diode for subsequent analysis.

**Optogenetic stimulation**. Laser stimulation was delivered through a fiber optic cord (200 μm diameter) using a laser diode fiber light source (Doric Lenses). Light intensity was calibrated and wavelength-corrected using the Power Meter Bundle with the PM100D Console and S130C Slim Photodiode Sensorlight (Thorlabs). Every pacing stimulation (including 8 Hz) was performed using 5 ms square pulses. To

apply scrambled light stimulation, we used an Arduino microcontroller to generate a white noise oscillation signal directly fed into the analog input of the laser diode driver. To perform randomly selected frequency stimulation, we used a standard 5 s ON, 5 s OFF protocol, but for each stimulation epoch, we used an Arduino microcontroller to randomly select a stimulation frequency in the theta band. When applying optogenetic stimulation during behavior, a loose piece of heat shrink tubing was fitted around the junction between the patch cord and the mouse ferrule implant to limit visible light emission. Light intensities are expressed as nominal power, as measured at the tip of the fiber optic implant–cord assembly (before surgical placement), and corrected for the appropriate wavelength.

### Behavioral assays

**Sequential tone linear track.** Mice were water scheduled and could access water only for 2 h per day from 6–8 pm. To disentangle spatial, temporal, and distance codes, we build a 134 cm long linear track using medium gray Lego® bricks, which allows easy modifications and implementations without the need to permanently modify the structure of the maze. Pyroelectric sensors were placed at each end of the linear track and were connected to an Arduino microcontroller. Each detection triggered a new tone in sequence, indicating the progress to reward delivery. The following tones were used and delivered using a piezo speaker: 1 s beeps at 2000 Hz, 250 ms beeps at 3000 Hz, and a continuous tone at 4000 Hz. When the last, continuous tone was triggered, a reward (10% sucrose in water) was delivered at the starting end of the linear track in the cap of a 15 mL Falcon tube. Changes in running direction before triggering the next tone were considered errors and did not trigger reward delivery. Performance was measured as the number of correct trials (with no error) divided by the number of total trials.

**Novel object place recognition.** On the first day, mice were allowed to freely explore a $45 \times 45$ cm dark gray open field that contained visual cues (large white horizontal and vertical gratings) on its walls for 10 min. On the second day, two identical objects (helping hand base) were presented for 10 min. On the third day, mice were allowed to explore the same open field while the location of one of the two objects was displaced. To control for potential spatial preference biases, both the initial, as well as the displaced position of objects, were randomized. Mice have attributed a random testing order that was kept identical throughout the three days of testing. Behavior was recorded with a video camera (Logitech) and analyzed offline. Behavioral analysis was performed blind to the genotype and treatment. Object explorations were defined as epochs where mice have their nose within 1 cm of an object. RI was computed as the following:

$$RI = \frac{t_{novel}}{t_{novel} + t_{familiar}} \tag{1}$$

where $t_{novel}$ and $t_{familiar}$ represent the time exploring novel and familiar objects, respectively. This metric was also computed in the sample phase for the prospectively novel (displaced) object to assess potential pre-existing preferential biases.

**Automated delayed non-match to sample task.** Mice were water-scheduled and trained in a continuous T-maze to a delayed non-match to sample task. Briefly, each trial was divided into two phases: sample and test. In the sample phase, one arm was blocked and mice were forced to explore the opposite arm, where they received a 50 µL of 10% sucrose water reward. After completing the sample phase, mice were subjected to a delay (10 s) in the starting compartment. Then, during the test phase, both arms could be explored, but only the opposite (unexplored) arm is baited so that mice had to alternate locations between a sample and test phases. Mice were subjected to 10 trials

(sample + test) per day, for 10 consecutive days, and the daily success rate was calculated as the number of correct trials divided by the total number of trials.

### Post-mortem histological analyses

After completion of behavioral testing, mice were deeply anesthetized with a mixture of ketamine/xylazine/acepromazide (100, 16, 3 mg/kg, respectively, intraperitoneal injection) and perfused transcardially with 4% paraformaldehyde (PFA) in PBS. Brains were extracted and postfixed overnight in PFA at 4 °C and subsequently washed in PBS for an additional 24 h at 4 °C. Brains and sections were cryoprotected in a solution of 30% ethylene glycol, 30% glycerol, and 40% PBS until used. Each brain was then sectioned at 50 µm using a vibratome: every section was sequentially collected in 4 different 1.5 mL tubes to allow different analyses (electrode location, immunohistochemistry).

**Immunohistology.** Using one tube of collected brain sections (25% sampling) for each analysis, sections were washed for $3 \times 5$ min in PBS to remove the cryoprotective solution. Sections were incubated overnight with PGT (0.45% Gelatin and 0.25% Triton in PBS) at 4 °C. Next, slices were incubated with primary antibodies: either 1:200 goat anti-choline-acetyl-transferase from Millipore or 1:500 mouse anti-PV monoclonal IgG1 from Sigma-Aldrich in PGT at room temperature for 48 h or 2 h respectively. Following washes of 10, 20, and 30 min, sections were then incubated with secondary antibodies [1:2000 donkey anti-goat coupled with Alexa 488 or 1:500 goat anti-mouse IgG1 coupled to Alexa 488 (Life Technologies)] in PGT for 45 min. Following 10, 20, and 30 min washes in PBS, sections were then mounted on glass slides and permanently coverslipped with Fluoromount mounting medium that contained DAPI. Only mice with the histologically confirmed placement of implants were included in this study. For GRIN lenses, the surface of the lens had to be <100 µm above *stratum pyramidale* and GCaMP6f expression was validated using fluorescence microscopy. Electrophysiological implants had to include at least one microelectrode in CA1 *stratum radiatum* or *stratum pyramidale*. Finally, the tips of fiber optics had to be within 100 µm of the MS region, and proper construct expression was assessed using fluorescence microscopy.

### Data analysis

Except for analyses of SWRs and the explicit impact of locomotion on physiological recordings, electrophysiological and calcium imaging analyses were performed only on periods of locomotion (>2 cm s$^{-1}$ in the open field; >5 cm s$^{-1}$ on the linear track).

**Automatized tracking of behavior.** To extract information about the position, velocity, and head direction of mice, we used DeepLabCut[70,71]. Briefly, we trained a model to detect mice's ears, nose, body, and tail base. Head direction was estimated using the angle between either each ear or the nose and body, depending on measurement availability. Location data was interpolated to calcium imaging sampling frequency using linear interpolation. Velocity was extracted by computing $\Delta d / \Delta t$ where $d$ is the distance and $t$ time and subsequently smoothed the result by applying a gaussian filter with sigma = 33 ms to remove detection artifacts. Velocity signals were used to identify periods of locomotor activity and compute place, time, and distance modulation of activity specifically for those periods.

**Calcium imaging analysis.** Calcium imaging data analysis was performed using MATLAB 2020a and Python 3.8.4. Video recordings were analyzed using the Miniscope Analysis pipeline (https://github.com/etterguillaume/MiniscopeAnalysis). Briefly, rigid (rotation and translation) motion correction was applied using NoRMCorre[72], and videos were spatially downsampled (3×) before concatenation. Calcium traces were extracted using CNMFe[51] using the following

parameters: gSig = 3 pixels (width of gaussian kernel), gSiz = 20 pixels (approximate neuron diameter), background_model = 'ring', spatial_algorithm = 'hals', min_corr = 0.8 (minimum pixel correlation threshold), min_PNR = 8 (minimum peak-to-noise ratio threshold).

Raw calcium traces were filtered to remove high-frequency fluctuations and binarized: briefly, neurons were considered active when normalized calcium signal amplitude exceeded two standard deviations, and the first-order derivative was above 0 (see ref. [52] for additional details on the methodology[52]). To extract neurons tuning to specific variables, location, time, and distance were binned (location: 3 cm bins; time: 1 s bins; distance: 3 cm bins). From binarized signals, we computed the marginal likelihood of cells being active $P(A)$ that we use as a proxy for neuronal activity. More importantly, we then derive activity likelihood or the 'probability of being active given the state of a variable' $P(A \mid S_i)$ using binned variables:

$$P(A \mid S_i) = \frac{\text{time active while in bin}i}{\text{time in state}} \qquad (2)$$

where $A$ is the activity level (0 or 1) of a given neuron, when being in bin $i$ of a given state $S$. These values were smoothed using a Gaussian kernel with $\sigma$ = 5, 10, or 15 for spatial, temporal, and distance tuning curves, respectively.

To identify the place, time, and distance of cells, we computed the MI between neuronal activity and the bin $i$ of a given state $S$ using the following:

$$MI = \sum_{i=1}^{M} \sum_{j=1}^{2} P(S_i \cap A_j) \times \log_2 \left( P \frac{(S_i \cap A_j)}{P(S_i) \times P(A_j)} \right) \qquad (3)$$

where $M$ is the total number of possible behavioral states, $P(S_i \cap A_j)$ is the joint probability of the animal being in bin $i$ concurrently with activity level $j$ (0 or 1). To assess the significance of the obtained MI value, we then generated 1000 shuffled surrogates using random circular permutations. We chose circular permutations because they remove the temporal relationship between neuronal activity and behavior, while still preserving the temporal structure of calcium transients and thus lead to more conservative results (as opposed to complete randomization of every data point, which inflates the significance value of results). Because shuffled surrogates were not systematically normally distributed, we used a non-parametric approach where the p-value ($p_N$) corresponds to the number of data points from the shuffled distribution that are greater than the actual data for each bin, divided by the number of permutations[52,73].

To determine the modulation of cells by optogenetic stimulations, we used a similar approach but computed the MI between neuronal activity and the binarized stimulation signals (thus treated as a behavioral state). The same circular shuffling procedure was then used to extract statistical significance.

**Naive Bayes decoder**. To further evaluate the significance of neuronal activity at the population level, we trained a naive Bayesian decoder with a uniform prior to predicting spatiotemporal variables from binarized neuronal activity. The posterior probability density function, which is the probability that an animal is in a particular bin $S_i$, given the neural activity $A$ is calculated using the following equation:

$$P(S_i \mid A) = \frac{P(A \mid S_i) \times P(S_i)}{P(A)} \qquad (4)$$

where $P(S|A)$ is the posterior probability distribution of states given neuronal activity. Using only epochs with velocity >5 cm s$^{-1}$, a training dataset was generated using 90% of the data. The remaining 10% of the data was used for testing. Decoding error was calculated using 50 bootstrapped surrogates and a pool of 160 cells using randomly

chosen data points with replacement. Every neuron is assumed to be independent of each other, which in practice is not the case and leads to greater reconstruction errors, but decreases computational time. The population posterior probability was derived from the following equation:

$$P(S \mid A) = \prod_{k=1}^{N} \frac{P(A_k \mid S) \times P(S)}{P(A_k)} \qquad (5)$$

with $N$ corresponding to the number of neurons used. Finally, the decoded data point corresponds to the maximum *a posteriori* (MAP):

$$\hat{y} = \text{argmax} \exp \left[ \sum_{k=1}^{N} \log \left( 1 + \frac{P(A_k \mid S) \times P(S)}{P(A_k)} \right) - 1 \right] \qquad (6)$$

The exponential-logarithm trick is used to prevent underflow when multiplying a large number of small probabilities, which is a correct approximation since the posterior is a smooth convex function.

Temporal filtering was applied to posterior probabilities using:

$$P(S \mid A_k(t)) = \prod_{k=1}^{N} \prod_{t=1}^{T} \frac{P(A_k(t) \mid S) \times P(S)}{P(A_k(t))} \qquad (7)$$

where $t$ is a given timestep, and $P(S \mid A_k(t))$ the *a posteriori* distribution of states at timestep $t$ for neuron $k$. The number of past timesteps used to estimate the mouse state at time $t$ is determined by $T$, which we set as 2 s unless noted otherwise. On the linear track, decoding error was assessed as:

$$\text{decoding error} = |\text{decoded state} - \text{actual state}| \qquad (8)$$

In order to compare decoding scores across animals, we z-scored decoding errors using the following equation:

$$z = \frac{\bar{x} - \bar{s}}{\sigma(x)} \qquad (9)$$

where $\bar{x}$ is the average actual decoding error, $\bar{s}$ is the average decoding error using shuffled surrogates, and $\sigma(x)$ is the standard deviation of the actual decoding error.

**Tracking cells over multiple days**. Neurons were tracked over multiple days using CellReg[74]: https://github.com/zivlab/CellReg (v1.5.3). Briefly, spatial footprints were aligned using rigid alignment to correct for rotations and translations. After alignment, we considered candidate sets of cells to be the same neuron if their maximal distance was <12 μm, and used the modeled spatial correlation threshold (usually in the range 0.6–0.8) to determine the identity of cell pairs across days. Finally, we assessed the stability of the spatial representation using pairwise field correlation (Pearson correlation of tuning curves).

**Electrophysiological analysis**. Electrophysiological data analysis was performed using MATLAB 2020a using both the signal processing as well as the wavelet toolbox. Wavelet convolution was applied to LFP signals using complex Morlet wavelets ('cmor1–1.5' in MATLAB) when both time- and frequency-domain accuracy was required. Moving window Fourier convolution (2 s window in the theta band, 5 s window in the gamma band, 10 ms moving steps) was used when frequency-domain accuracy was privileged over time-domain accuracy (e.g., to plot dominant frequency when pacing using optogenetics). Analysis of power spectral densities was performed when mice were running at 5 cm s$^{-1}$ or above, unless described otherwise.

**Oscillation strength**. OS was computed as the ratio of cumulative power spectral density around the peak oscillation frequency ±1 Hz to

the cumulative band power in the theta (4–12 Hz) band. This metric becomes 1 when all power spectral density falls within the peak oscillation frequency (e.g. 8 Hz if stimulating at that frequency).

**SWRs detection and analysis.** To monitor SWRs, mice were allowed to freely explore an open field for 10 min while recording. Stimulations (scrambled or 8 Hz) were performed with a 5 s ON, 5 s OFF paradigm. Only periods of quiet restfulness were considered for analysis. To this end, we computed the $z$-scored ratio of theta/delta power after filtering for each frequency band, performing a Hilbert transform, and only considered periods where the resulting value was under 0. To detect SWRs, we filtered LFP signals in the 150–250 Hz frequency band and subsequently $z$-scored. Ripples were detected using the findpeaks function in MATLAB, with the following parameters: threshold = 4 sd, minpeakwidth = 0 s, minpeakdistance = 0.03 s.

## Statistics

Unless stated otherwise, all data are presented as mean ± standard error of the mean (SEM) and statistical test details are described in the corresponding results. The distribution normality of each group was assessed using Shapiro–Wilk normality test and parametric tests were used only when distributions were found normal (non-parametric tests are described where applicable). 1ANOVA: one-way ANOVA; 2ANOVA: two-way ANOVA; RM-ANOVA: repeated measures ANOVA. $p \leq 0.05$ was considered statistically significant. *$p \leq 0.05$; **$p \leq 0.01$; ***$p \leq 0.001$, ****$p \leq 0.0001$. The effect size of 1ANOVA was assessed with $\eta^2$ using the sum of squares (SS):

$$\eta^2 = \frac{SS_{\text{between}}}{SS_{\text{total}}} \tag{10}$$

Effect size of 2ANOVA and RM-ANOVA was assessed with partial $\eta^2$ using:

$$\eta^2{}_p = \frac{SS_{\text{effect}}}{SS_{\text{effect}} + SS_{\text{error}}} \tag{11}$$

## Reporting summary

Further information on research design is available in the Nature Portfolio Reporting Summary linked to this article.

## Data availability

The processed dataset generated in this study is publicly available at https://osf.io/78wez/ or via a request to the corresponding authors. Source data for all main text and Supplementary Figures can be found in the Source Data file Etter_NC2022_Source_Data.xlsx provided with this article. Source data are provided in this paper.

## Code availability

All source codes used in the current study (analysis of calcium imaging and electrophysiological data) are available along with instructions[75]. Extraction of calcium imaging data was done using https://github.com/etterguillaume/MiniscopeAnalysis (v1.0). This extraction pipeline leverages motion correction from NoRMCorre https://github.com/flatironinstitute/NoRMCorre (v0.1.1) as well as CNMFe video processing https://github.com/zhoupc/CNMF_E (v1.1.2) specifically for V3 miniscopes.

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

## Acknowledgements

We thank Ke Cui for helping to maintain the colony and perfusing mice; Mark P. Brandon, Alexandra T. Keinath, and James E. Carmichael for comments on the paper. This work was supported by funding from the Canadian Institutes for Health Research (CIHR) Foundation Program FDN-148478, the Natural Sciences and Engineering Research Council of Canada (NSERC) Discovery Grant RGPIN-2020-06717, and a Tier 1 Canada Research Chair to S.W. S.V.D.V. was supported by a Vanier Canada Graduate Scholarship and the Richard H. Tomlinson Doctoral Fellowship. J.C. was supported by a Healthy Brains, Healthy Lives (HBHL) Graduate Student Fellowship and a McGill Integrated Program in Neuroscience (IPN) Internal Student Award. The funders had no role in study design, data collection and analysis, decision to publish, or preparation of the paper.

## Author contributions

G.E. and S.W. designed the study. G.E. performed and analyzed in vivo experiments. S.V.D.V. performed and analyzed both behavioral as well as immunohistological experiments. J.C. performed surgeries for the mice used for controlling cross-talk between calcium imaging and optogenetics. G.E. and S.W. wrote the paper with inputs from all collaborators.

## Competing interests

The authors declare no competing interests.
