## [Peer Review File · Nature Communications]

Optogenetic frequency scrambling of hippocampal theta oscillations dissociates working memory retrieval from hippocampal spatiotemporal codesREVIEWER COMMENTS

Reviewer #1 (Remarks to the Author):

Etter et al. examine object place memory and delayed alternation in mice under conditions wherein gabaergic septal inputs to the hippocampus are optogenetically stimulated either at 8 Hz (mimicking the theta-frequency rhythm generated in hippocampus by septal inputs) or with an irregular pattern (designed to eliminate the theta-frequency rhythm). The authors also provide the same stimulus protocols while imaging Ca transients in hippocampal neurons during back and forth shuttling on a linear track. The authors claims include the presence of place, time, and distance neurons in hippocampus and that these tuning dynamics are not altered by septal stimulation. However, memory performance was impacted by stimulation, providing the basis for a dissociation of spatial memory from tuning to spatial information in hippocampus. The work is thorough in examining local field potentials to test for the impact of stimulation protocols on temporal dynamics of neural activity in hippocampus. The finding of “time” and “distance” tuning in freely behaving animals (neither head restrained nor confined to a treadmill) is potentially impactful and novel, but the authors do not strongly support the existence of such tuning and refrain from deep consideration of the issue. Some of the effects of stimulation on memory performance are subtle and the effects on hippocampal neuron tuning to place, time, and distance seem strong even if they are statistically insignificant. Thus, the main findings are not convincingly supported by the data.

1) Elapsed distance and time encoding in hippocampal activity has, to my knowledge, only been observed in restrained animals (head-restraint or treadmill). This has led to considerable skepticism concerning the importance of time and distance tuning under normal circumstances. The authors present evidence that such tuning may occur in freely moving animals, but it is not sufficient to make a strong case (and a strong case could be the most interesting aspect of the work). I suggest the authors show many more examples (in main or supplemental figures), demonstrate reliability across trials, and show the distribution of time and distance peaks (not clearly discernable from the figures). If there is to be a claim to time and distance tuning, the strength of this must be shown beyond the information theoretic approach (which provides little evidence of tuning strength) and beyond the decoding approach (for which the level of error is high).

2) While the extensive testing of optogenetic stimulation on locomotor speed is appreciated, the result is largely a control and could be relegated to supplemental figures. The significant, but extremely weak ($r \sim 0.11$) correlation between locomotor speed and theta frequency is overstated with respect to significance (as speed explains only 1% of variability in frequency).

3) Some of the effects on behavior (figure 8f,g) are small. More data would potentially provide more credibility to the finding.

4) Figure 1n,q,t – The decoding errors for place (40 cm), time (15 sec), and distance (175 cm) are all very high, suggesting that the Ca transients recorded are not well related to the variables under study. In figure 6, the non-significant changes in place, time, and distance tuning are rather large (approximate doublings), suggesting too low power for analysis. In fact, one can imagine more data yielding weak effects on behavior but strong effects on tuning (opposite the manuscript's message).

Reviewer #2 (Remarks to the Author):

In this manuscript by Etter et al by the Williams group, the authors investigate the impact of medial septal (MS) manipulations on hippocampal neural dynamics and working memory performance by using optogenetics manipulations of septal PV GABAergic neurons and miniscope calcium imaging. They report that optogenetic frequency scrambling of MS GABAergic PV+ cells' activity abolishes theta oscillation in the hippocampal area CA1 and modulates the activity of a subpopulation of CA1 neurons. Such perturbation leads to impaired memory retrieval but not encoding in a delayed non-match to sample task and object location recognition task. The authors report that scrambled MS-PV stimulation is not associated with disrupted encoding of place, time, distance, or multiplexed information. The authors conclude that theta oscillation plays a specific role in supporting working memory retrieval, but it is not essential for spatiotemporal coding in the hippocampus. Overall, this is an interesting and elegant study. The authors report overall negative results on many physiological and biological outcome measures of hippocampal function upon MS-PV-based manipulations of hippocampal theta. While this is potentially interesting, it also warrants a more convincing demonstration of the efficacy of the manipulation used in the study. I have some other comments related to the experiments and analyses that should be addressed by the authors before a further recommendation can be made.

Major

1. The MS-PV scrambling is an elegant way to impair theta without grossly changing the overall activity levels of this input to the hippocampus. However, only a relatively small fraction (~15%) of MS-PV cells were transfected and presumably manipulated (line 156 and Fig. 2). While it is difficult to assess the magnitude of reduction in baseline theta power by scrambled MS-PV stimulation, it appears that it is around 30-40% (Fig. 2i) and ~25% or less during GRIN Lens imaging (Fig. 3e). This partial effect may limit the interpretation of the negative result on hippocampal neural dynamics and behavioral outcomes in the study.

2. Relatedly, the definition of 'baseline' theta is unclear – is this measured during locomotion bouts? Can the authors show that locomotion-related theta is abolished by MS-PV manipulation in the task? Some related analyses are provided in Fig. 7, but the magnitude of reduction in theta associated with locomotion is not quantified.

3. The analysis of hippocampal neural population dynamics is carried out on a subset of the total population of place-modulated, time-modulated, or distance-modulated cells representing ~9%, 1.8%, and 1.2% of the population, respectively, while analyses mostly ignore the majority of conjunctive neurons. The coding quality of these subsets is not very robust with decoding error even for their preferred modality is relatively large (Fig. 1 l-t), and the effect size of decoding accuracy over shuffle is small. In addition, these already small fractions are further subdivided when the effect of MS-PV manipulations is analyzed over days, which yields a very small number of cells across mice. For example, it seems that altogether n=10 distance-cells were recorded, which is probably 1-2 cells per mouse (?). So, this part of the study appears underpowered, and the conclusions that can be drawn are somewhat limited. This point should be addressed by the authors.

4. Figure 5: This reviewer is confused by the experimental design. Why are coding fractions and stability are compared only across days? Wouldn't we want to know, first and foremost, the acute effect of MS-PV manipulation on coding properties within-session/day? Based on what is presently shown, it seems possible that theta scrambling affects within-session but not across-day coding, which would go against the authors' argument that theta is important only for retrieval and not for encoding.

6. Fig. 8: It is unclear why MS-PV manipulation was not performed during the sample phase in the object displacement task. Without this experiment, it seems unwarranted to conclude theta is needed only for retrieval.

REVIEWER COMMENTS

Reviewer #1 (Remarks to the Author):

Etter et al. examine object place memory and delayed alternation in mice under conditions wherein gabaergic septal inputs to the hippocampus are optogenetically stimulated either at 8 Hz (mimicking the theta-frequency rhythm generated in hippocampus by septal inputs) or with an irregular pattern (designed to eliminate the theta-frequency rhythm). The authors also provide the same stimulus protocols while imaging Ca transients in hippocampal neurons during back and forth shuttling on a linear track. The authors claims include the presence of place, time, and distance neurons in hippocampus and that these tuning dynamics are not altered by septal stimulation. However, memory performance was impacted by stimulation, providing the basis for a dissociation of spatial memory from tuning to spatial information in hippocampus. The work is thorough in examining local field potentials to test for the impact of stimulation protocols on temporal dynamics of neural activity in hippocampus.

The finding of “time” and “distance” tuning in freely behaving animals (neither head restrained nor confined to a treadmill) is potentially impactful and novel, but the authors do not strongly support the existence of such tuning and refrain from deep consideration of the issue. Some of the effects of stimulation on memory performance are subtle and the effects on hippocampal neuron tuning to place, time, and distance seem strong even if they are statistically insignificant. Thus, the main findings are not convincingly supported by the data.

1) Elapsed distance and time encoding in hippocampal activity has, to my knowledge, only been observed in restrained animals (head-restraint or treadmill). This has led to considerable skepticism concerning the importance of time and distance tuning under normal circumstances. The authors present evidence that such tuning may occur in freely moving animals, but it is not sufficient to make a strong case (and a strong case could be the most interesting aspect of the work). I suggest the authors show many more examples (in main or supplemental figures), demonstrate reliability across trials, and show the distribution of time and distance peaks (not clearly discernable from the figures). If there is to be a claim to time and distance tuning, the strength of this must be shown beyond the information theoretic approach (which provides little evidence of tuning strength) and beyond the decoding approach (for which the level of error is high).

We thank reviewer #1 for the time invested in reviewing our work. We agree that there has been little evidence of temporal coding in freely moving animals beyond treadmill paradigms and expanding on this part of the study would benefit the manuscript. We have now added a supplemental figure with additional examples of neurons modulated by space, time or distance (supplemental fig. 2, line 127). To tie these more descriptive examples to our information theoretic approach, we ranked neurons by MI value and show both significant and non-significant examples. In preliminary analyses, we computed split-half stability for each neuron, but found that this method tends to include artifacts (e.g. if a single transient divides the recording in the two epochs used to compute stability). We solve this issue by expressing tuning curves in activity likelihood (i.e. likelihood of 0.7 corresponds to a neuron firing 70% of the time in a particular state and thus expresses a measure of stability). Importantly, some tuning curves can be associated with high MI values, but are not significantly different from chance - such examples are shown in the new supplemental figure 2. Note that spatially modulated cells activity likelihood rarely crosses 0.5 - this is due to the fact that CA1 pyramidal cells are known to only fire with one specific direction on a linear track. Additionally, we also improved our decoding approach. Details have been included in our response to comment #4.

2) While the extensive testing of optogenetic stimulation on locomotor speed is appreciated, the result is largely a control and could be relegated to supplemental figures. The significant, but extremely weak ($r \sim 0.11$) correlation between locomotor speed and theta frequency is overstated with respect to significance (as speed explains only 1% of variability in frequency).

We agree with reviewer #1 that this experiment is only used as a control for effects observed during behavioral testing. We have now moved this figure to supplemental figure 6. Additionally, and as requested by reviewer #2, we now include analyses of theta optogenetic control depending on locomotor speed, and

find that we can abolish/pace theta at any speed, including restfulness periods (supplemental fig. 3, lines 178-185) which we find more meaningful considering the claims proposed in our study.

3) Some of the effects on behavior (figure 8f,g) are small. More data would potentially provide more credibility to the finding.

We agree that the effects shown on behavior are small, while significant. Our analysis does include both inter-subject (YFP vs Chrimson mice) and intra-subject variability (treatment), which could have been misinterpreted in our first version, where data for YFP mice was plotted separately. We now plot both groups conjointly. Additionally, we have computed effect size for treatment, which amounts to $\eta^2_p = 0.196$ when applying stimulations in the retrieval phase, and $\eta^2_p = 0.109$ when applying stimulations in the delay period, which corresponds to high (>0.14) and medium (>0.06) effect sizes (Cohen, 1988). Effect size for groups in the NOPR task was $\eta^2_p = 0.306$. For this reason, we did not include additional mice to previous conditions, and these effect sizes are now reported (lines 368,375). However, we did add a new group to the NOPR task in the new version of our manuscript (stimulations during encoding phase).

4) Figure 1n,q,t – The decoding errors for place (40 cm), time (15 sec), and distance (175 cm) are all very high, suggesting that the Ca transients recorded are not well related to the variables under study. In figure 6, the non-significant changes in place, time, and distance tuning are rather large (approximate doublings), suggesting too low power for analysis. In fact, one can imagine more data yielding weak effects on behavior but strong effects on tuning (opposite the manuscript's message).

As pointed out by reviewer #2, decoding errors were high in the first version of our manuscript. In our new version, our decoder now includes (1) temporal filtering of posterior probabilities, (2) Gaussian smoothing of tuning curves, (3) 90/10 train/test ratio which were missing in the first version of the manuscript. We used parameters suggested by other groups for reference (Doron et al., 2022; Rubin et al., 2019; Sheintuch et al., 2020) which are now detailed in the method section (lines 756, 758, 771), as well as a slightly larger bootstrap sample size ($n=160$ cells instead of 100). Using these parameters, we were able to decrease decoding errors by half for all variables. Now, instead of comparing spatial vs non-spatial cells only for their coding properties on all variables (including time and distance) we were now able to compare temporal vs non-temporal, as well as distance vs non-distance coding neurons (fig. 1n,q,t, right panels). In figure 6, we computed the z-scored values using these new decoding parameters and still recapitulated that our stimulations had no effects of spatiotemporal codes. We computed effect size for one-way ANOVAs and found $\eta^2 = 0.29$ (spatial), 0.07 (temporal), and 0.09 (distance) which corresponds to large, medium, and medium effect sizes respectively (Cohen, 1988). These results are now reported lines 289-292.

Reviewer #2 (Remarks to the Author):

In this manuscript by Etter et al by the Williams group, the authors investigate the impact of medial septal (MS) manipulations on hippocampal neural dynamics and working memory performance by using optogenetics manipulations of septal PV GABAergic neurons and miniscope calcium imaging. They report that optogenetic frequency scrambling of MS GABAergic PV+ cells' activity abolishes theta oscillation in the hippocampal area CA1 and modulates the activity of a subpopulation of CA1 neurons. Such perturbation leads to impaired memory retrieval but not encoding in a delayed non-match to sample task and object location recognition task. The authors report that scrambled MS-PV stimulation is not associated with disrupted encoding of place, time, distance, or multiplexed information. The authors conclude that theta oscillation plays a specific role in supporting working memory retrieval, but it is not essential for spatiotemporal coding in the hippocampus. Overall, this is an interesting and elegant study. The authors report overall negative results on many physiological and biological outcome measures of hippocampal function upon MS-PV-based manipulations of hippocampal theta. While this is potentially interesting, it also warrants a more convincing demonstration of the efficacy of the manipulation used in the study. I have some other comments related to the experiments and analyses that should be addressed by the authors before a further recommendation can be made.

Major

1. *The MS-PV scrambling is an elegant way to impair theta without grossly changing the overall activity levels of this input to the hippocampus. However, only a relatively small fraction (~15%) of MS-PV cells were transfected and presumably manipulated (line 156 and Fig. 2). While it is difficult to assess the magnitude of reduction in baseline theta power by scrambled MS-PV stimulation, it appears that it is around 30-40% (Fig, 2i) and ~25% or less during GRIN Lens imaging (Fig. 3e). This partial effect may limit the interpretation of the negative result on hippocampal neural dynamics and behavioral outcomes in the study. We are thankful to reviewer #2 for taking the time to review our manuscript. We understand the concern with potentially low magnitude effects of our optogenetic stimulations based on the figures of the original manuscript. While power spectra derived from fourier transforms convey that theta is abolished under scrambled stimulations in the form of a flat curve (fig. 2h, top panels), we agree that non-zero power spectral density (PSD) values could suggest to the reader that some theta oscillations remain. To clarify this part of our study, we added a new panel (fig2i) where we assessed oscillation strength (clearly described in the Methods section, lines 806-810). Briefly, oscillation strength was computed as the ratio of cumulative power spectral density around the peak oscillation frequency ± 1 Hz to the cumulative band power in the theta (4 - 12 Hz) band. Importantly, we computed oscillation strength on a pure sine wave (which results in values close to 1, which is more easily interpretable than unbounded PSD values). Since, oscillation strength is not zero even in complete absence of theta we also compute oscillation strength on white noise (random signal with same amplitude as LFP trace) to obtain a floor value for oscillation strength. Using this method, we found that scrambled stimulation significantly decreased oscillation strength compared to baseline, and these values are in the same range (not significantly different) as the ones computed on white noise. These results are now presented at lines 167-172 and fig. 2i.*

2. *Relatedly, the definition of 'baseline' theta is unclear – is this measured during locomotion bouts? Can the authors show that locomotion-related theta is abolished by MS-PV manipulation in the task? Some related analyses are provided in Fig. 7, but the magnitude of reduction in theta associated with locomotion is not quantified.*

We apologize for the lack of clarity on this part, especially considering the important role of locomotion on theta rhythms generation. In addition to adding details in the Methods section (lines 690-692), we now have additional experiments to assess the effect of optogenetic stimulations on theta oscillations depending on locomotor speed (supplemental fig. 3). We found that we were able to systematically abolish theta (with scrambled stimulations) or drive theta (with 8 hz stimulations) regardless of ongoing locomotor speed i.e. and both periods of restfulness as well as running epochs (supplemental fig. 3; lines 178-185). We are very thankful to reviewer #2 for suggesting a more meaningful analysis approach.

3. *The analysis of hippocampal neural population dynamics is carried out on a subset of the total population of place-modulated, time-modulated, or distance-modulated cells representing ~9%, 1.8%, and 1.2% of the population, respectively, while analyses mostly ignore the majority of conjunctive neurons. The coding quality of these subsets is not very robust with decoding error even for their preferred modality is relatively large (Fig. 1 l-t), and the effect size of decoding accuracy over shuffle is small. In addition, these already small fractions are further subdivided when the effect of MS-PV manipulations is analyzed over days, which yields a very small number of cells across mice. For example, it seems that altogether n=10 distance-cells were recorded, which is probably 1-2 cells per mouse (?). So, this part of the study appears underpowered, and the conclusions that can be drawn are somewhat limited. This point should be addressed by the authors.*

We agree with the reviewer that our previous decoding analysis yielded high prediction error, and included a limited amount of cells. We have addressed the high prediction errors in response to reviewer #1. In brief, gaussian smoothing of tuning curves was previously omitted but is now included, temporal filtering of posterior probabilities has now been implemented, and training/test set have been changed from 50/50 to 90/10 (we based our parameters on references including Doron et al., 2022; Rubin et al., 2019; Sheintuch et al., 2020). We have now updated figure 1m,n,p,q,s,t and lines 138-149. We also agree that focusing all analyses on neurons that only encode one variable limits the power of statistical analyses, and does not

recapitulate previous findings as most studies do not explicitly exclude neurons that encode more than one variable. We have now included analyses of decoding error for conjunctive neurons (supplemental fig. 5, line 274). Additionally, we are now making mention of this sampling limitation in our discussion (lines 455-458).

4. Figure 5: This reviewer is confused by the experimental design. Why are coding fractions and stability are compared only across days? Wouldn't we want to know, first and foremost, the acute effect of MS-PV manipulation on coding properties within-session/day? Based on what is presently shown, it seems possible that theta scrambling affects within-session but not across-day coding, which would go against the authors' argument that theta is important only for retrieval and not for encoding.

We agree with reviewer #2 that this analysis is a limitation of our study. Namely, to sample temporal and distance cells, mice must travel on the tone-driven linear track 4x more than on a regular track. When applying chronic stimulations, this resulted in insufficient sampling in order to determine whether phasic stimulations have a significant effect. Firstly, we have now added this limitation to our discussion (line 448-4455). To further address reviewer #2's comment, we have now performed chronic (5s ON, 5s OFF) stimulations in the same open field where we ran our novel object task. In these conditions, we were able to quantify spatial tuning (albeit not time nor distance tuning). We computed the stability of rate maps during baseline vs stimulated epochs, so as to obtain a measure of MS PV manipulation of coding properties within-session. In these conditions, we found that spatial tuning remained stable during optogenetic stimulations (fig. 4f,g; line 246).

6. Fig. 8: It is unclear why MS-PV manipulation was not performed during the sample phase in the object displacement task. Without this experiment, it seems unwarranted to conclude theta is needed only for retrieval.

We agree with the statement of the reviewer that optogenetic stimulations of the medial septum during encoding are required to control for the specificity of effects observed during memory retrieval, as described in our original version of the manuscript. To answer this question, we have now added new experiments for animals that underwent scrambled or 8 Hz stimulation during the sample, but not the test phase of the novel object place recognition task (fig. 7b, green and yellow data points). We found that scrambled stimulation replicated our findings in the T-maze, that is that theta oscillations are only required during retrieval of spatial memory, not encoding. On the other hand, we were surprised to find that 8 Hz stimulations did impair memory encoding, which does not completely recapitulate our findings observed in the T-maze task (line 347).

While this disparity is likely explained by the fact that the DNMTS and NOPR tasks assess different aspects of spatial memory (spatial working memory and spatial recognition memory, respectively) we also set out to further understand the mechanisms of 8 Hz effects on memory encoding. We recorded LFP in dorsal CA1 using two electrodes 1 mm apart from each other along the septotemporal axis. When applying 8 Hz stimulations, we found that optogenetic light pulses robustly dictated theta phase (supplemental fig. 7a,b). Moreover, cross-correlation increased dramatically between the two sites (supplemental fig. 7c,d). We proposed that these extreme conditions which are not normally observed under physiological conditions could be responsible for memory impairments (line 376-385). We now discuss these results and the potential impact of theta phase manipulations at lines 442-447.

REVIEWERS' COMMENTS

Reviewer #1 (Remarks to the Author):

The authors have responded as best possible, given the data, to the issues raised in the first review. However, the issues and concerns raised in the first review remain. First, there is little to evidence the presence of "time" or "distance" neurons in the present dataset and analyses. In particular, the intra-session reliability of firing across specific times or distances is not given; with such small numbers of such cells, the intra-session reliability could be shown for all. Again, the claim of time and distance cells in freely behaving animals is not to be taken lightly, yet the evidence presented is not convincing. There are suprisingly few "spatial" cells. All 3 types of cell (spatial, time, distance) seem remarkably unreliable across sessions (especially visually in figure 5 e-g). With such noisy signals, it is not that surprising to see a lack of effect. Nevertheless, it is still the fact that the impact of scrambled stimulation on firing of neurons is larger than the significant effects on behavior.

Reviewer #2 (Remarks to the Author):

The authors did an excellent job in their revision. I have no further requests.

Reviewer #1 (Remarks to the Author):

The authors have responded as best possible, given the data, to the issues raised in the first review. However, the issues and concerns raised in the first review remain. First, there is little to evidence the presence of "time" or "distance" neurons in the present dataset and analyses. In particular, the intra-session reliability of firing across specific times or distances is not given; with such small numbers of such cells, the intra-session reliability could be shown for all. Again, the claim of time and distance cells in freely behaving animals is not to be taken lightly, yet the evidence presented is not convincing. There are suprisingly few "spatial" cells. All 3 types of cell (spatial, time, distance) seem remarkably unreliable across sessions (especially visually in figure 5 e-g). With such noisy signals, it is not that surprising to see a lack of effect. Nevertheless, it is still the fact that the impact of scrambled stimulation on firing of neurons is larger than the significant effects on behavior.

We thank reviewer #2 for taking the time to review our manuscript. Concerning the evidence of time and distance neurons, stability metrics are reported under the form of a probability for which we show several examples in supplemental fig. 2 (a value of 1 representing perfect stability). Importantly, our definition of spatial, temporal, and distance cells is conservative i.e. they are only considered when encoding one variable specifically. When considering all spatial cells (spatial cells 'proper' + conjunctive spatial cells cells), we find ~30% of spatially tuned neurons, which is comparable to other calcium imaging studies (e.g. Ziv et al., 2013; Rubin et al., 2019).

Reviewer #2 (Remarks to the Author):

The authors did an excellent job in their revision. I have no further requests.

We thank reviewer #2 for the comments on the manuscript